# Decentralized Instruction Tuning: Conflict-Aware Splitting and Weight Merging

**Minsik Choi** [* 1 2 ‡]    **Geewook Kim** [* 1 3 †]

## Abstract

Instruction tuning aligns large language models, including multimodal ones, with diverse user intents, but scaling to heterogeneous mixtures is hindered by gradient interference and bandwidth-heavy synchronization. We ask whether these two bottlenecks can be addressed jointly by training parts of the mixture independently and reconciling them once in parameter space. We develop a local quadratic theory inside a shared flat basin that yields three results: weight merging produces a curvature-weighted variance reduction; PCA-aligned conflict splitting maximizes this gain along high-curvature directions; and merging additionally acts as spectral filtering with implicit norm regularization. These results directly motivate **MERIT**, a decentralized merge-ready instruction-tuning pipeline that estimates dataset-level gradient conflicts, partitions the mixture along the top PCA conflict axes, fine-tunes each partition independently with no inter-partition communication, and merges once via token-weighted averaging. On Qwen2.5-VL-3B with 136 Vision-FLAN tasks, MERIT improves the 8-benchmark average from 54.3 (joint training) to 57.0. The same recipe scales to a 7B model on a 1.6M-example, 176-source mixture—matching or exceeding centralized joint training with minimal cost overhead—and transfers to text-only FLAN. Our code is available at https://github.com/naver-ai/merit.

## 1. Introduction

The modern recipe for building capable foundation models relies on extensive post-training, in particular large-scale instruction tuning over a diverse mixture of tasks. This

---

[*]Equal contribution [‡]This work was conducted during Minsik Choi's internship at NAVER Cloud. [1] NAVER Cloud AI [2] Korea University [3] KAIST AI . Correspondence to: [†]Geewook Kim <gwkim.rsrch@gmail.com>.

*Proceedings of the 43rd International Conference on Machine Learning*, Seoul, South Korea. PMLR 306, 2026. Copyright 2026 by the author(s).

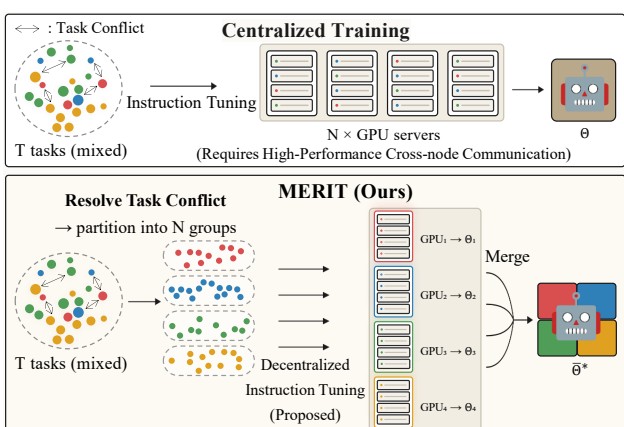

*Figure 1.* **Centralized training vs. MERIT.** Centralized tuning synchronizes conflicting tasks across a tightly-coupled cluster (top); MERIT partitions the mixture by conflict, fine-tunes each group independently, and merges once into $\bar{\theta}$ (bottom).

step is particularly critical for multimodal large language models (MLLMs) (Liu et al., 2024a;b), where much of the practical multimodal behavior is acquired after language pretraining through vision–language alignment and heterogeneous instruction data spanning perception, reasoning, OCR and document understanding, diagram comprehension, and safety (Xu et al., 2024b; Laurençon et al., 2024). As a result, practitioners keep scaling instruction mixtures to achieve robust, product-level capability (Hu et al., 2024; Wang et al., 2024; Kim & Seo, 2024; Bai et al., 2025; Li et al., 2025; Gemma Team et al., 2025; Chen et al., 2024).

However, the prevailing approach, centralized joint training on the entire mixture, faces two bottlenecks. On the optimization side, heterogeneous datasets induce conflicting gradients, leading to negative transfer and stiff dynamics that constrain the learning rate and slow progress (analyzed in Section 3). Classical multi-task corrections that manipulate task-wise gradients (Chen et al., 2018; Yu et al., 2020; Liu et al., 2021) become infeasible at the scale of over a hundred tasks and billions of parameters (Appendix E.1), leaving mixture-ratio curation (Laurençon et al., 2024) as the main recourse. On the systems side, joint training relies on frequent gradient synchronization (e.g., all-reduce), effectively requiring tightly-coupled clusters with high-bandwidth interconnects; this assumption breaks in fragmented compute environments such as heterogeneous GPU pools, geo-

distributed clusters, and cloud spot instances, where communication is a dominant cost or a hard constraint.

These two bottlenecks are *coupled*: as mixtures grow more heterogeneous, optimization becomes more sensitive to dataset interference, yet mitigating such interference during training demands fine-grained, synchronized training signals. When synchronization is unavailable, practitioners fall back to coarse mixing heuristics, leaving interference largely unaddressed.

This calls for an alternative: instead of forcing all datasets to share a single synchronized trajectory, can we train parts of the mixture independently and reconcile them once in parameter space? Weight-space averaging and model soups (Wortsman et al., 2022; Jang et al., 2024) suggest that when fine-tuning runs start from the same checkpoint and remain within a connected low-loss region, one-shot parameter averaging can yield a single model that is often better than its constituents. Crucially, this weight-space compatibility is common in post-training, where fine-tuning typically stays within a shared flat basin around a strong initialization. We term such a checkpoint a *merge-ready initialization* (empirically verified via linear mode connectivity, displacement, and perturbation diagnostics in Appendix B) and ask: how should we split a large instruction mixture so that independent training remains mergeable and merging systematically reduces interference? Existing soup-style methods average models trained on the same data with different randomness; our setting instead requires merging models trained on disjoint partitions of a heterogeneous mixture, where the choice of split directly determines merging quality.

We answer this question with **MERIT** (**Me**rge-**R**eady **I**nstruction **T**uning), a decentralized instruction-tuning pipeline built on two moves at a merge-ready initialization—**split** the mixture along the dominant gradient-conflict axes identified by PCA, then **fine-tune** each partition independently with no cross-partition communication—followed by a single token-weighted merge (Figure 1). Structurally, the split separates conflicting updates across branches; operationally, the pipeline reduces post-training to a problem that fragmented, bandwidth-constrained hardware can solve. A local quadratic analysis inside a shared flat basin (Section 3) makes this concrete: the merging gain is a curvature-weighted variance reduction, PCA-aligned splitting maximizes it, and averaging implicitly regularizes toward $\theta^{(0)}$ while acting, conceptually, as spectral filtering.

Our main contributions are:

- **Theoretical framework.** A local quadratic analysis inside a shared flat basin derives three results: merging yields a curvature-weighted variance reduction; PCA-aligned splitting maximizes this gain, with the advantage growing with the Hessian spectral gap; and weight averaging acts as implicit norm regularization and, conceptually, spectral filtering (Section 3).

- **Decentralized instruction-tuning pipeline.** MERIT estimates dataset-level gradient conflicts from a small calibration set, partitions the mixture along top-$r$ PCA conflict axes at a merge-ready initialization, trains $K{=}2^r$ branches fully independently, and merges once via token-weighted averaging—no cross-partition gradient communication during fine-tuning.

- **Consistent empirical gains at scale.** On Qwen2.5-VL-3B with 136 Vision-FLAN tasks (Xu et al., 2024b), MERIT improves the 8-benchmark average from **54.3** → **57.0** under identical token budgets; on a Qwen2.5-VL-7B build with a 1.6M-example 176-source mixture, MERIT matches or exceeds centralized joint training across three independent seeds with minimal wall-clock overhead (Section 6), and the same recipe transfers to a text-only FLAN setting (Wei et al., 2022).

## 2. Background and Related Work

**Model Merging and Loss-Landscape Connectivity.** Weight-space model merging averages multiple checkpoints into a single model and is motivated by loss-landscape connectivity (Garipov et al., 2018; Draxler et al., 2018) and flat-minima interpretations (Hochreiter & Schmidhuber, 1997; Izmailov et al., 2018). Model soups and Model Stock show that simple averaging over independently fine-tuned models can be effective (Wortsman et al., 2022; Jang et al., 2024), while follow-up work explores more robust merging rules, including curvature-aware variants and conflict-mitigating heuristics (e.g., sign alignment) (Matena & Raffel, 2022; Yadav et al., 2023; Yu et al., 2024a). Most prior work merges models trained on the same dataset/objective (different seeds or hyperparameter settings), where averaging mainly reduces run-to-run variance. To disentangle such generic averaging gains from our contributions, our experiments include soup-style baselines (uniform averaging of multiple runs on the full mixture) and random dataset partitioning followed by averaging. MERIT instead targets heterogeneous instruction mixtures and introduces an a priori conflict-aware split so that models trained on disjoint subsets remain mergeable and benefit from averaging.

**Federated and Decentralized Optimization.** Independent local training followed by averaging is reminiscent of FedAvg and Local SGD (McMahan et al., 2017; Stich, 2019), which periodically synchronize client models to optimize a shared global objective under siloed data and communication/privacy constraints (Kairouz et al., 2021). Recent work extends federated learning to LLM fine-tuning via communication-efficient strategies such as shared-

randomness gradient compression (Zelikman et al., 2023), seed-based full-parameter tuning (Qin et al., 2024), and forward-pass perturbation (Xu et al., 2024a), while low-rank (Hyeon-Woo et al., 2022) and personalization techniques (Qin et al., 2023; 2025) further reduce per-round cost. However, these methods still require iterative synchronization rounds. One-shot federated learning (Li et al., 2021; Diao et al., 2023; Li et al., 2024c; Huang & Shu, 2025) removes iterative communication by aggregating after a single local-training round, sharing MERIT's single-merge structure. A key distinction is that FL methods typically assume fixed, pre-assigned data partitions dictated by privacy or ownership constraints, whereas MERIT assumes centrally available data and optimizes the partition itself to maximize merging quality via gradient-conflict analysis. This active partitioning, a degree of freedom absent in the FL setting, is the primary driver of MERIT's performance gains.

**Task Interference in Multi-Task and Instruction Tuning.** Negative transfer from conflicting gradients is a central challenge in multi-task learning; gradient reweighting/projection methods mitigate interference but typically require synchronized access to per-task gradients during training (Chen et al., 2018; Yu et al., 2020; Liu et al., 2021). Instruction tuning is likewise sensitive to mixture composition (Longpre et al., 2023; Sanh et al., 2022). In multi-modal post-training, Vision-FLAN reports conflicts between diverse short-answer tasks and verbose conversational responses (Xu et al., 2024b). More broadly, alignment datasets can encode competing objectives (e.g., helpfulness vs. harmlessness) (Askell et al., 2021). In the multimodal setting, recent studies show that vision-language adaptation can impact model safety (Lee et al., 2025a), motivating behavior-alignment pipelines such as RLHF-V (Yu et al., 2024b). MERIT complements these lines by separating conflicting datasets before training and reconciling them once in parameter space, avoiding step-level interference accumulation without requiring synchronized per-step control.

**Data Curation and Mixture Design for (M)LLMs.** Recent (M)LLM pipelines build large instruction corpora by curating and mixing heterogeneous datasets with careful ratio tuning (Zhou et al., 2023; Liu et al., 2023; Laurençon et al., 2024). As new datasets are added or priorities shift, mixture design becomes an iterative bottleneck. MERIT complements curation with a reusable decomposition primitive: it estimates dataset interactions at a shared initialization, enables communication-free parallel fine-tuning during training, and merges once at the end.

**Positioning of Our Work.** MERIT sits at the intersection of model merging, decentralized training, and instruction-mixture learning. Unlike post-hoc merging methods, MERIT introduces an a priori conflict-aware dataset

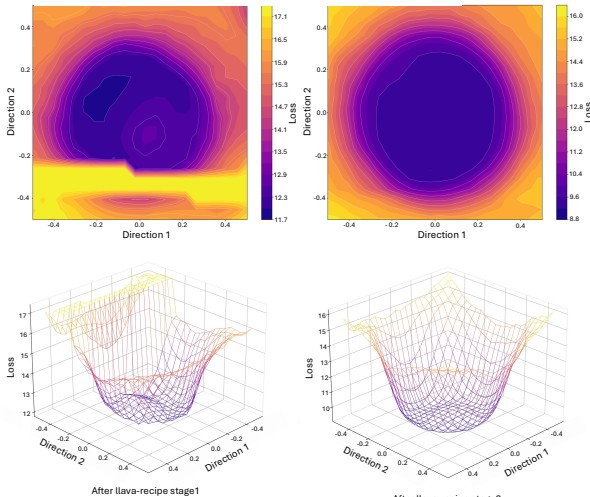

*Figure 2.* Local loss surfaces before and after basin preparation (e.g., LLaVA Stage 2). The merge-ready initialization $\theta^{(0)}$ resides in a flat, connected region (right), where independently fine-tuned checkpoints remain within the same basin. Our analysis operates within this flat-basin regime, yielding three key implications that directly motivate MERIT's algorithm design.

split to make simple averaging effective under disjoint heterogeneous data. Unlike gradient-level multi-task methods, MERIT moves conflict handling before training and removes step-level synchronization. Unlike federated averaging/local SGD, MERIT chooses partitions to reduce dataset interference and is analyzed under a flat-basin merging theory rather than as an approximation to centralized training on a single objective.

## 3. Theoretical Framework

We present a theoretical framework explaining why weight merging improves optimization and generalization when fine-tuned checkpoints remain confined to a common flat basin (Figure 2). Our analysis yields three key implications, each experimentally verified in Sections 5–6, that directly motivate MERIT (Section 4): merging is never worse than the weighted average of individual losses in the quadratic model, with improvement governed by curvature-weighted variance; conflict-aware PCA splitting maximizes this gain over random partitioning, with the advantage growing with the curvature gap; and merging acts as implicit norm regularization and, conceptually, as curvature-weighted spectral filtering, improving both generalization and optimization dynamics. Formal assumptions and complete proofs are deferred to Appendix F.

**Setting.** Let $\theta \in \mathbb{R}^d$ denote the model parameters and $L(\theta)$ the population loss. We consider $K \geq 2$ checkpoints $\theta_1, \ldots, \theta_K$, each fine-tuned on a different dataset group

from a shared *merge-ready initialization* $\theta^{(0)}$—a checkpoint from which independently fine-tuned models remain in a connected low-loss region. We describe how to obtain such an initialization in Section 4. We fix a reference point $\theta^\star$ within the basin, define displacements $\delta_i = \theta_i - \theta^\star$ (for the first-order analysis below we take $\theta^\star = \theta^{(0)}$; the gain $\mathcal{G}_{\mathrm{var}}$ is invariant to this choice), and let $H = \nabla^2 L(\theta^\star) \succeq 0$ denote the local Hessian. The merged checkpoint is $\bar{\theta}_w = \sum_i w_i \theta_i$ with weights $w_i \geq 0$ summing to one, with uniform averaging recovered as a special case of the token-weighted averaging used by MERIT.

**Merging yields curvature-weighted variance reduction.** Under a local quadratic approximation $L(\theta) \approx L(\theta^\star) + \frac{1}{2}(\theta - \theta^\star)^\top H (\theta - \theta^\star)$, weighted averaging yields a deterministic gain:

$$\underbrace{\sum_{i=1}^{K} w_i L(\theta_i) - L(\bar{\theta}_w)}_{\text{merging gain } \mathcal{G}_{\mathrm{var}}} = \frac{1}{2} \sum_\ell \lambda_\ell \, \mathrm{Var}_w\big(u_\ell^\top \delta_i\big) \geq 0$$

$$(1)$$

where $\lambda_\ell, u_\ell$ are eigenvalue–eigenvector pairs of $H$, and $\mathrm{Var}_w(x_i) := \sum_i w_i x_i^2 - (\sum_i w_i x_i)^2$ denotes the $w$-weighted variance over $\{x_i\}_{i=1}^{K}$.

**Theorem 3.1** (Merging gain in a flat basin). *Under a local quadratic model with $H \succeq 0$, $L(\bar{\theta}_w) \leq \sum_i w_i L(\theta_i)$. The inequality is strict whenever the displacements $\{\delta_i\}$ are not identical modulo the nullspace of $H$. The gain $\mathcal{G}_{\mathrm{var}}$ is maximized when checkpoint dispersion concentrates along the dominant eigendirections of $H$.*

The gain therefore depends on *where* the dispersion lies: variance along high-curvature directions dominates, while variance in flat subspaces contributes little. This motivates splitting strategies that concentrate inter-group updates along curvature-bearing directions, the role of conflict-aware PCA in MERIT.

**Conflict-aware splitting maximizes the merging gain.** The above result shows that $\mathcal{G}_{\mathrm{var}}$ grows with curvature-aligned dispersion. We now show that PCA-based splitting achieves this alignment.

Under a first-order fine-tuning approximation from $\theta^{(0)}$, the displacement of branch $k$ is $\delta_k \approx -\eta \, \bar{g}_k$, where $\bar{g}_k$ is the mean gradient of datasets assigned to group $k$. The merging gain for two groups becomes

$$\mathcal{G}_{\mathrm{var}} = \frac{\eta^2}{8} (\bar{g}_1 - \bar{g}_2)^\top H (\bar{g}_1 - \bar{g}_2).$$

Since the gradient of dataset $t$ at $\theta^{(0)}$ satisfies $g_t = -H\Delta_t$ (where $\Delta_t$ is the centered per-dataset optimum), the gain is governed by $H^3$-weighted interactions among the per-dataset optima. This curvature amplification means that

PCA on gradient conflicts preferentially identifies high-curvature disagreement axes.

**Proposition 3.2** (PCA-aligned splitting maximizes $\mathcal{G}_{\mathrm{var}}$). *In a tractable linear-quadratic model (Appendix F.4), PCA-aligned partitioning selects the high-curvature disagreement axis and provably outperforms random partitioning: it attains the maximum gain among all balanced partitions in the $T{=}4$, $d{=}2$ instance (Proposition F.6), and for general $T$ exceeds random partitioning in expectation under a spectral-concentration condition (Proposition F.11). The advantage grows with the spectral gap $\lambda_1/\lambda_2$, and the gain scales as $\lambda_1^3$—curvature enters twice through gradient conflict ($g_t \propto H\Delta_t$) and once through the Hessian-weighted gain formula. MERIT's recursive splitting along the top-$r$ PCA axes ($K{=}2^r$) accumulates such gains across $r$ orthogonal high-curvature directions, consistent with the monotonic $1D \to 2D \to 3D$ improvements observed in Section 5.*

Although PCA is applied to dataset-level gradient similarities rather than the Hessian directly, gradient disagreement at a shared initialization provides a first-order proxy for curvature-sensitive update directions under local smoothness. Appendix F.2 formalizes the connection between cosine-similarity and raw-gradient PCA, and Appendix F.3 empirically confirms across two MLLM scales that PCA conflict directions partially align with the top Hessian eigenvectors (far above a Gaussian random baseline), supporting this proxy interpretation.

**Merging as spectral filtering with implicit norm regularization.** The preceding results establish that conflict-aware merging *reduces loss*. We now show it also *improves optimization dynamics* and *regularizes parameters*.

*Spectral filtering (conceptual).* Joint training on a heterogeneous mixture is constrained by stability along the stiffest curvature direction: the learning rate must satisfy $\eta < 2/\lambda_{\max}$, forcing slow progress along flatter dimensions when the condition number $\kappa = \lambda_{\max}/\lambda_{\min}$ is large. PCA-structured splitting and merging instead drive $U^\top(\bar{\theta}_w - \theta^\star) \approx 0$, where $U$ spans the dominant curvature subspace, suppressing high-curvature error components. This acts as approximate spectral filtering, lowering the effective condition number to $\kappa_{\mathrm{eff}} \approx \lambda_{\perp,\max}/\lambda_{\min} \ll \kappa$ (Appendix F.1.1), trading the slow, stability-constrained descent of joint training for a one-shot neutralization of high-curvature conflicts; we verify the predicted stability benefit empirically under an SGD learning-rate sweep.

*Implicit norm regularization.* Weight averaging additionally contracts the merged model toward the shared initialization. By convexity of the squared norm,

$$\|\bar{\theta}_w - \theta^{(0)}\|^2 \leq \sum_i w_i \|\theta_i - \theta^{(0)}\|^2, \qquad (2)$$

*Table 1.* Merge-readiness diagnostics (Qwen2.5-VL-3B, MERIT-2D, $K=4$ branches). The merged model stays closer to $\theta^{(0)}$ while carrying higher training loss, yet generalizes better.

| Epoch | Displacement $\|\cdot -\theta^{(0)}\|_2$ | | | Training loss | | |
|---|---|---|---|---|---|---|
| | Joint | Merged | Ratio | Joint | Merged | Gap |
| 0.5 | 13.73 | 5.65 | 2.43× | 0.709 | 1.198 | +0.489 |
| 1.0 | 19.73 | 7.50 | 2.63× | 0.560 | 1.172 | +0.611 |
| 2.0 | 28.15 | 10.11 | 2.78× | 0.370 | 1.167 | +0.797 |
| 6.0 | 34.61 | 11.87 | 2.92× | 0.064 | 1.330 | +1.266 |

with strict inequality whenever checkpoints differ. This contraction yields a smaller distance-based complexity term under PAC-Bayes generalization bounds, providing a principled account of why MERIT can improve generalization even when individual branch training losses are not lower than joint training (see Table 1).

**Empirical verification of merge-readiness.** The analysis above rests on two assumptions—branches remaining in a single flat basin and averaging contracting displacements along high-curvature directions—which we verify directly on Qwen2.5-VL-3B's $K=4$ branches (MERIT-2D) through four diagnostics (full details in Appendix B). First, along all 6 pairwise and 4 branch-to-merged linear interpolation paths, the loss barrier $\max_\alpha[L(\theta(\alpha)) - ((1-\alpha)L(0) + \alpha L(1))]$ is *exactly zero*: every path stays at or below linear interpolation, the strongest possible form of linear mode connectivity. Second, consistent with Eq. (2), the merged model remains 2.4–2.9× closer to $\theta^{(0)}$ than the jointly trained model throughout training, with the ratio widening monotonically (see Table 1). Third, despite carrying a substantially *higher* training loss on the full mixture ($+0.49$ to $+1.27$ across epochs), the merged model achieves better held-out performance—the classic signature of implicit regularization, and precisely the behavior predicted by the norm-contraction argument above (formalized via PAC-Bayes in Appendix F.5). Finally, isotropic Gaussian weight perturbations ($\sigma \in \{0.01, 0.05, 0.1\}$) produce consistently smaller loss increases on the merged model than on the jointly trained one, confirming a flatter surrounding landscape (Appendix B.2).

## 4. Proposed Method

**Overview.** MERIT is a decentralized instruction-tuning pipeline that turns a heterogeneous dataset mixture into $K=2^r$ conflict-aware partitions, fine-tunes each partition independently from a shared merge-ready initialization $\theta^{(0)}$, and merges the resulting checkpoints via token-weighted averaging. The pipeline has five stages (Algorithm 1): (i) dataset-level gradient conflict estimation, (ii) PCA-based decomposition of the conflict structure, (iii) balanced partitioning along the dominant PCA axes, (iv) communication-free

---

**Algorithm 1** MERIT: Conflict-Aware Dataset Partitioning and Weight Merging

**Require:** Merge-ready initialization $\theta^{(0)}$; datasets $\{\mathcal{D}_t\}_{t=1}^T$ with sample counts $\{s_t\}_{t=1}^T$ and token budgets $\{n_t\}_{t=1}^T$; PCA dimension $r$.
**Ensure:** Merged model $\bar{\theta}$.
1: ▷ *Step 1: Gradient conflict estimation at $\theta^{(0)}$.*
2: **for** $t = 1, \ldots, T$ **do**
3:     Compute $g_t$ at $\theta^{(0)}$ under identical training settings (backbone, trainable-parameter subset, gradient-estimation budget); set $\tilde{g}_t \leftarrow g_t/\|g_t\|$.
4: **end for**
5: Form $C \in \mathbb{R}^{T \times T}$ with $C_{ij} = \langle \tilde{g}_i, \tilde{g}_j \rangle$.
6: ▷ *Step 2: PCA-based conflict decomposition.*
7: Apply (column-centered) PCA to $C$ and obtain the top-$r$ PCA embedding $z_t \in \mathbb{R}^r$ for each $t$.
8: ▷ *Step 3: Balanced conflict-aware partitioning.*
9: Recursively split $\{1, \ldots, T\}$ along the $r$ PCA axes via sample-balanced medians (weights $s_t$) into $K = 2^r$ disjoint groups $\{\mathcal{G}_k\}_{k=1}^K$, balancing per-group sample counts $\sum_{t \in \mathcal{G}_k} s_t$; let $N_k = \sum_{t \in \mathcal{G}_k} n_t$ denote the per-group token budget (with $\sum_{k=1}^K N_k = \sum_{t=1}^T n_t$).
10: ▷ *Step 4: Communication-free group-wise training.*
11: **for** $k = 1, \ldots, K$ **do**
12:     Train $\theta_k$ from $\theta^{(0)}$ on $\bigcup_{t \in \mathcal{G}_k} \mathcal{D}_t$ using budgets $\{n_t\}_{t \in \mathcal{G}_k}$.
13: **end for**
14: ▷ *Step 5: Token-weighted parameter-space merging.*
15: **return** $\bar{\theta} = \sum_{k=1}^K w_k \theta_k$ with $w_k = N_k / \sum_{j=1}^K N_j$.

---

branch training, and (v) weight merging.

**Merge-ready initialization.** A prerequisite for MERIT is a merge-ready initialization $\theta^{(0)}$: a checkpoint from which independently fine-tuned models remain in a connected low-loss region and therefore admit effective one-shot merging. In our multimodal experiments, $\theta^{(0)}$ is an already instruction-tuned MLLM checkpoint (e.g., released Qwen2.5-VL); in text-only settings a pretrained LLM is sufficient. Merge-readiness is an empirical property of the chosen initialization; comprehensive diagnostics (linear mode connectivity, weight displacement, and perturbation robustness) are provided in Appendix B.

### 4.1. Dataset-Level Gradient Conflict Estimation

MERIT quantifies dataset-level interference by comparing the directions of per-dataset gradients at $\theta^{(0)}$. For each dataset $t$, we estimate a representative gradient $g_t$ by averaging gradients over a small calibration set (up to 200 examples per dataset), using identical model and trainable-

parameter settings. We then normalize each gradient and construct the cosine similarity matrix,

$$C_{ij} := \frac{\langle g_i, g_j \rangle}{\|g_i\| \, \|g_j\|},$$

where high values indicate aligned updates and low or negative values indicate conflicting updates under joint optimization. Cosine-based gradient alignment is a standard tool for characterizing task interactions in multitask and continual learning (Lopez-Paz & Ranzato, 2017; Yu et al., 2020; Ilharco et al., 2023; Lei et al., 2026).

On a 100-task subset of Vision-FLAN, the mean cosine similarity between the $n{=}200$ calibration gradient and the full 1,000-sample reference gradient reaches $0.847$ (std $0.106$), with diminishing returns beyond $n{=}200$ (Appendix C.4); a qualitative t-SNE view of these dataset-level gradients is provided in Appendix C.1. To keep preprocessing cheap at scale, we subsample one gradient entry every $s$ parameters (default $s{=}5$, 20% retained); the resulting cosine similarities match the full-gradient baseline with Pearson/Spearman correlations above $0.98$ (Appendix E). Finally, the matrix can be reused across collections: extending an existing $T$-dataset collection with $m$ new datasets needs only $O(Tm)$ additional similarity computations rather than recomputing the full $O((T{+}m)^2)$ matrix, followed by a negligible PCA update.

### 4.2. PCA-Based Conflict Decomposition

We extract the dominant conflict structure of $C$ via PCA, yielding an $r$-dimensional embedding $z_t \in \mathbb{R}^r$ per dataset that captures the principal patterns of agreement and disagreement among dataset-level gradients. We consider $r \in \{1, 2, 3\}$, giving the 1D, 2D, and 3D variants of MERIT (with $K{=}2^r \in \{2, 4, 8\}$ branches).

We use cosine-similarity PCA as the default: it is scale-invariant and recovers the same leading eigenspace as raw-gradient PCA under gradient-norm concentration, which we verify empirically across our dataset mixtures (Appendix F.2).

### 4.3. Conflict-Aware and Balanced Dataset Partitioning

Using the PCA embedding, MERIT partitions datasets into disjoint groups so that datasets with similar projections are assigned to the same group while datasets with opposing projections are separated. This spreads group-wise updates across distinct, conflicting directions from $\theta^{(0)}$.

A practical challenge is group imbalance: naive thresholding along PCA coordinates can yield groups with highly unequal data volumes. MERIT instead performs recursive 50/50 splits along PCA coordinates using sample-balanced medians, where each dataset carries weight $s_t$ equal to its

sample count. This balances group sizes without sacrificing separation along dominant conflict axes. A comparison with distance-based clustering (K-means on the same gradient representation) is provided in Appendix C.3.

### 4.4. Communication-Free Branch Training

After partitioning, MERIT fine-tunes one model per group, all initialized from $\theta^{(0)}$ and sharing the same backbone, trainable-parameter subset, and hyperparameters; the only difference is which datasets each group sees. No cross-group communication occurs during training, so branches can run in parallel on disjoint hardware.

For a fair comparison to centralized joint training, MERIT matches the total budget. Each dataset $t$ is assigned to exactly one group $k$ and contributes its full budget $n_t$ in that branch, so the per-group token count $N_k := \sum_{t \in \mathcal{G}_k} n_t$ satisfies $\sum_k N_k = \sum_t n_t$, matching the joint-training baseline.

### 4.5. Token-Weighted Merging

The resulting branch checkpoints $\theta_1, \ldots, \theta_K$ are merged in a single pass via token-weighted averaging:

$$\bar{\theta} = \sum_{k=1}^{K} w_k \, \theta_k, \qquad w_k = \frac{N_k}{\sum_{j=1}^{K} N_j}.$$

When per-group token budgets are exactly balanced, this reduces to uniform averaging; under our sample-balanced split the token budgets are only approximately balanced, so the merge is in general non-uniform.

## 5. Experiments

**Setup.** We evaluate MERIT in a controlled 3B study on Qwen2.5-VL-3B (Bai et al., 2025) with 136 Vision-FLAN tasks (Xu et al., 2024b), reporting 8 multimodal benchmarks grouped into four categories (General MCQA, User Preference & Fluency, Text-Rich VQA, Image Reasoning). Baselines include centralized joint training at 0.5/1/2 epochs, random partitioning (2/4/8 groups), conflict-induced splitting, and uniform model soups. Full experimental details (training recipe, hyperparameters, calibration set, per-benchmark protocols, and statistical significance tests) are provided in Appendix A. Large-scale 7B results and text-only transfer are reported in Section 6.

### 5.1. Overall Performance

Table 2 summarizes the controlled 3B study. Every MERIT variant outperforms single-epoch joint training on the 8-benchmark average; even random partitioning surpasses Joint 1 ep, consistent with Theorem 3.1 predicting a deterministic merging gain within a shared flat basin. The best variant, MERIT-3D, improves over the primary Joint

*Table 2.* Controlled comparison of multimodal post-training strategies across diverse benchmarks. **Avg.** denotes the mean score over all benchmarks. **Bold** indicates the best result in each column, and underline indicates the second-best result in each column. Unless otherwise noted, all numbers are averaged over 3 independent runs; Joint training (1 ep) and MERIT-3D are averaged over 5 seeds as our primary comparison. Statistical significance tests for Joint training (1 ep) vs. MERIT-3D are reported in Appendix C.2.1.

| Method | General MCQA | | User Preference & Fluency | | Text-Rich VQA | | Image Reasoning | | Avg. |
|---|---|---|---|---|---|---|---|---|---|
| | SeedBench | MMBench | LLaVA-W | MMVet | TextVQA | AI2D | MathVista | MMMU | |
| Base model | 66.8 | 79.7 | **53.2** | 34.0 | 61.2 | **63.8** | 29.6 | 41.2 | 53.7 |
| Joint training (0.5 ep) | 67.4 | 79.4 | 40.2 | 33.1 | 67.2 | 60.9 | 31.4 | 41.6 | 52.7 |
| Joint training (1 ep) | 69.2 | 80.5 | 41.9 | 36.4 | 68.0 | 62.6 | 34.2 | 41.9 | 54.3 |
| Joint training (2 ep) | 70.0 | **81.4** | 42.8 | 37.6 | 63.4 | 62.5 | 36.5 | **43.0** | 54.7 |
| Random (2 groups) | 69.4 | 80.1 | 44.5 | 34.7 | 70.4 | 62.7 | 34.0 | 41.2 | 54.6 |
| Random (4 groups) | 70.4 | 81.0 | 40.6 | 34.7 | 70.4 | 63.1 | 34.0 | 40.8 | 54.4 |
| Random (8 groups) | 69.5 | 79.9 | 42.2 | 35.0 | 73.7 | 61.7 | 33.5 | 40.5 | 54.5 |
| Conflict-induced (2 groups) | 70.7 | 80.6 | 42.6 | 35.4 | 70.0 | 62.9 | 34.4 | 42.3 | 54.9 |
| Uniform soup (2) | 70.2 | 81.1 | 45.0 | 35.3 | 68.9 | 63.4 | **36.8** | 42.2 | 55.4 |
| Uniform soup (3) | 70.1 | 81.1 | 42.3 | 36.3 | 68.8 | 63.1 | 35.8 | 42.5 | 55.0 |
| Uniform soup (4) | 70.2 | 81.1 | 41.8 | 36.3 | 68.4 | 63.4 | 35.9 | 42.2 | 54.9 |
| **MERIT (Proposed, 1D split, 2 groups)** | **71.0** | 80.0 | 43.1 | 35.0 | 72.4 | 62.1 | 36.5 | 41.4 | 55.2 |
| **MERIT (Proposed, 2D split, 4 groups)** | 70.8 | 78.4 | 47.4 | 36.6 | 74.1 | 61.5 | 36.0 | 40.7 | 55.7 |
| **MERIT (Proposed, 3D split, 8 groups)** | 70.5 | 80.1 | 52.0 | **37.7** | **75.2** | 62.5 | 35.4 | 42.7 | **57.0** |

*Table 3.* Comparison over LLaVA-Series on diverse MLLM benchmarks under two 7B base models. We report each of the eight benchmarks together with their overall mean (**Avg.**, computed only when all eight scores are available). For each base model, we compare further full fine-tuning via centralized Joint FFT against MERIT under a matched training budget; MERIT uses the 2D split with $K=4$ groups. **Bold** marks the better of Joint FFT vs. MERIT within the same base. For the 0.7M-base build we report the first seed, and its Joint FFT and MERIT are each validated over three independent seeds in Appendix C.2; the stronger 3.6M-base build is a single run.

| Model | Train Data | General MCQA | | User Preference & Fluency | | Text-Rich VQA | | Image Reasoning | | Avg. |
|---|---|---|---|---|---|---|---|---|---|---|
| | | SeedBench | MMBench | LLaVA-W | MMVet | TextVQA | AI2D | MathVista | MMMU | |
| LLaVA-7B | 0.6M | 37.0 | 38.7 | 57.2 | 25.5 | – | 48.3 | 25.4 | 34.1 | – |
| LLaVA-1.5-7B | 0.7M | 65.9 | 64.3 | 59.6 | 31.1 | 58.2 | 54.8 | 25.6 | 35.3 | 49.4 |
| LLaVA-1.5-13B | 0.7M | 68.2 | 67.7 | 66.1 | 36.1 | 61.3 | 59.5 | 27.7 | 33.6 | 52.5 |
| + Joint FFT (Xu et al., 2024b) | 0.7M + 0.2M | – | 69.8 | 38.5 | 33.4 | – | – | – | 34.4 | – |
| Base VLM 7B | 0.7M | 64.5 | 67.2 | 57.2 | 28.9 | 57.6 | 46.5 | 26.9 | 32.8 | 47.7 |
| + Joint FFT | 0.7M + 1.6M | 70.1 | **76.2** | 58.8 | 32.5 | **73.8** | **58.6** | 32.9 | **36.4** | 54.9 |
| **+ MERIT (Proposed, 2D)** | 0.7M + 1.6M | 70.6 | 75.7 | **59.2** | **35.1** | 72.4 | 58.4 | 35.4 | 36.1 | **55.4** |
| Scaled base VLM 7B | 3.6M | 69.8 | 74.5 | 67.1 | 34.0 | 72.4 | 69.2 | 39.4 | 45.1 | 58.9 |
| + Joint FFT | 3.6M + 1.6M | 71.3 | 75.3 | 50.2 | **41.5** | **80.2** | 71.6 | **49.7** | **47.4** | 60.9 |
| **+ MERIT (Proposed, 2D)** | 3.6M + 1.6M | **71.5** | **75.8** | 66.2 | 39.1 | 79.8 | **71.9** | 43.0 | 44.8 | **61.5** |

1 ep baseline by +2.7 without any cross-partition gradient communication during fine-tuning. The pattern replicates at 7B scale across three independent seeds, confirming consistency (Table 9 in Appendix C.2).

### 5.2. Conflict-Aware vs. Random Partitioning

A natural question is how much of MERIT's gain comes from its conflict-aware split, rather than from splitting and merging alone. To isolate this, we compare MERIT against random partitioning under the same number of groups, the same budget, and the same one-shot merge; the only difference is how datasets are assigned.

The gap is substantial. MERIT-3D improves over Random (8 groups) by +2.5 under identical budgets and the same

one-shot merging step, attributable purely to the choice of split. Consistent with Proposition 3.2, the advantage grows monotonically with the number of PCA dimensions, while random partitioning shows no such trend. Beyond random splits, MERIT also outperforms K-means clustering on the same gradient representations, indicating that aligning the partition with conflict directions matters more than simply grouping similar datasets (Appendix C.3).

## 6. Further Analyses and Discussions

### 6.1. Large-Scale Vision–Language Model Experiments

To test whether MERIT's split-and-merge gains persist as we scale both model size and data, we apply it to further

*Table 4.* Text-only benchmark results on Qwen2.5-3B with 66 FLAN tasks. **Bold** indicates the best result in each column, and underline indicates the second-best result in each column.

| Method | Knowledge QA | | Commonsense | | Text Inference | | Problem Solving | | Avg. |
|---|---|---|---|---|---|---|---|---|---|
| | MMLU | GPQA | HellaSwag | WinoGrande | BoolQ | XNLI | ARC-C | HumanEval | |
| Base model | 65.5 | 29.5 | 74.5 | 70.1 | 77.3 | **42.9** | 55.6 | 37.8 | 56.7 |
| Joint training (1 ep) | 65.8 | 30.1 | 74.0 | 69.9 | 83.9 | 41.9 | 55.5 | 39.7 | 57.6 |
| Joint training (2 ep) | **66.1** | 30.6 | **76.3** | 69.2 | 82.9 | 41.9 | **56.5** | **42.7** | 58.3 |
| Random (2 groups) | 65.8 | 29.5 | 74.4 | 70.3 | 85.3 | 42.4 | 55.1 | 42.4 | 58.2 |
| Random (4 groups) | 66.0 | 30.3 | 74.7 | 70.3 | 85.7 | 42.5 | 54.9 | 42.1 | 58.3 |
| Conflict-induced (2 groups) | 65.8 | 28.7 | 74.0 | 69.4 | 85.3 | 42.2 | 53.8 | 42.1 | 57.7 |
| Uniform soup (2) | 65.9 | 30.6 | 74.0 | 70.2 | 83.9 | 41.9 | 55.2 | 40.2 | 57.7 |
| Uniform soup (3) | 65.7 | **30.8** | 74.0 | 70.2 | 84.2 | 42.1 | 55.7 | 39.0 | 57.7 |
| **MERIT (Proposed, 1D split, 2 groups)** | 66.0 | 30.6 | 74.4 | 69.9 | **86.3** | 42.1 | 54.2 | 41.2 | 58.1 |
| **MERIT (Proposed, 2D split, 4 groups)** | **66.1** | 30.0 | 74.7 | **70.8** | 86.1 | 42.4 | 55.3 | 41.5 | **58.4** |

full fine-tuning (FFT) of a 7B vision–language model on a 1.6M-example mixture drawn from 176 sources (details in Appendix A.2). We run the same comparison on top of two base models of increasing strength, holding the 1.6M FFT mixture and the training budget identical across methods. Table 3 reports the 8-benchmark average for each build.

**Base VLM.** The first base follows a standard LLaVA-style recipe (feature alignment followed by a 0.7M-example instruction-tuning stage). Here MERIT improves the average over centralized Joint FFT ($54.9 \rightarrow 55.4$) while achieving a more balanced profile: it recovers the open-ended degradation that Joint FFT suffers on *User Preference & Fluency* (+2.6 on MMVet) and adds a clear gain on *Image Reasoning* (+2.5 on MathVista), at the cost of staying within roughly a point and a half on text-rich benchmarks. Across three independent runs (Appendix C.2), MERIT outperforms Joint FFT on the 8-benchmark average in every run, confirming the gain is robust to training randomness.

**Scaled base VLM.** A natural concern is that MERIT's gains might wash out once the base model is already strong. Recent recipes show that inserting a *high-quality knowledge learning* stage—training on large-scale high-quality image captioning data—substantially strengthens a VLM before instruction tuning (Li et al., 2024b; 2025). Following these recipes, we build a stronger base by adding this stage to the pipeline above: starting from feature alignment, we train on 2.9M image captioning samples (*stage 1.5*) and then on the 0.7M instruction-tuning data (*stage 2*), for 3.6M examples in total. This produces a markedly stronger starting point, on top of which we repeat the Joint FFT vs. MERIT comparison with the same 1.6M FFT mixture.

Even from this stronger initialization, MERIT again exceeds Joint FFT on the overall average ($60.9 \rightarrow 61.5$), though the headroom over the base is naturally smaller for both methods. The gap is concentrated on open-ended generation:

Joint FFT collapses LLaVA-Wild to short, generic answers ($67.1 \rightarrow 50.2$), whereas MERIT largely preserves the base model's open-ended quality (66.2; cf. the short-answer collapse analysis in Appendix D.1). This collapse is not specific to this run: the same pattern appears in our primary 3B comparison (Table 2), averaged over five seeds, where Joint training drops LLaVA-Wild from the base model's 53.2 to 41.9 while MERIT preserves it (52.0). On the more saturated perception and text-rich benchmarks the two remain comparable. On the reasoning-heavy benchmarks MERIT still improves over the base (MathVista $39.4 \rightarrow 43.0$) or holds it about flat (MMMU $45.1 \rightarrow 44.8$); Joint FFT reaches higher scores there, but in tandem with the open-ended collapse noted above, so the two differ mainly in capability profile while MERIT retains the better overall average.

**Takeaway.** Together, these results show that conflict-aware splitting followed by one-shot merging generalizes to 7B-scale post-training under both larger mixtures and stronger initializations.

### 6.2. Application to Text-only Instruction Tuning

We further validate MERIT on text-only instruction tuning using 66 FLAN (Wei et al., 2022) tasks with Qwen2.5-3B. Table 4 shows that MERIT-2D achieves the best average, exceeding 1-epoch joint training by +0.8 and matching or slightly exceeding 2-epoch joint training at half the budget, confirming that conflict-aware splitting generalizes beyond multimodal settings.

### 6.3. Efficiency Analysis

MERIT is designed for bandwidth-constrained, decentralized settings, eliminating step-level synchronization by independently fine-tuning branch models and merging them once. Per-task gradient manipulation methods such as PC-Grad (Yu et al., 2020) and GradNorm (Chen et al., 2018)

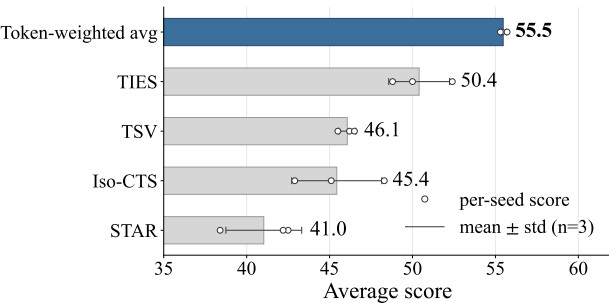

*Figure 3.* Post-hoc merging baselines applied to MERIT's 7B 2D branches ($K$=4). Each operator replaces MERIT's token-weighted averaging as a drop-in merge step on the same four branches. Bars show the mean over 3 seeds on the 8-benchmark suite; circles mark per-seed scores and error bars denote $\pm 1$ std. Token-weighted averaging outperforms all four alternatives; we attribute this to MERIT's branches being complementary by construction, so trimming- or orthogonalization-based operators can discard branch-specific content (Section 6.4).

are infeasible at our scale: PCGrad alone requires memory exceeding both our V100×8 and A100×8 systems at $T$=136 tasks on a 3B model (Appendix E.1). MERIT's only preprocessing cost is dataset-level gradient estimation; with the stride-$s$ subsampling introduced in Section 4, this cost is small and one-time.

At 3B scale, MERIT's parallel branch training and one-shot merge run only $\sim 24\%$ above single-epoch joint training on 8 V100 GPUs while achieving a substantially better average, and faster than two-epoch joint training; the one-time gradient-conflict preprocessing ($\sim$2h, amortizable across runs) is reported separately. At 7B scale this recurring overhead drops to $0.8\%$ on 8 A100 GPUs, with preprocessing an even smaller one-time fraction of total training (full breakdown in Appendix E).

### 6.4. Post-Hoc Merging Baselines

A natural question is whether MERIT's token-weighted averaging could be replaced with more sophisticated post-hoc merging operators. We evaluate four state-of-the-art alternatives (TIES (Yadav et al., 2023), STAR (Lee et al., 2025b), TSV (Gargiulo et al., 2025), and Iso-CTS (Marczak et al., 2025)) as drop-in replacements on MERIT's 7B 2D branches over three seeds. Token-weighted averaging consistently outperforms all four alternatives by 5–15 points on the 8-benchmark average (Figure 3). The reason is structural: these operators target redundancy among models specialized on the same task, but MERIT's branches are complementary by construction—each carries task-specific information the others lack. Trimming or orthogonalizing such signals discards content only one branch holds, whereas token-weighted averaging preserves all branch contributions (per-seed results in Appendix C.5).

## 7. Conclusion

We introduced MERIT, a decentralized instruction-tuning pipeline grounded in a local flat-basin quadratic analysis showing that weight merging yields curvature-weighted variance reduction, that PCA-based conflict-aware splitting maximizes this gain, and that merging acts as spectral filtering with implicit norm regularization. MERIT splits datasets before fine-tuning using gradient-conflict PCA, trains group-wise branch models without cross-group synchronization, and merges them once via token-weighted averaging. Experiments on multimodal and text-only instruction mixtures (Qwen2.5-VL-3B on Vision-FLAN and a 7B-scale setting with a 1.6M-example mixture) confirm these implications and show consistent improvements on the 8-benchmark average over centralized joint training while enabling communication-free parallel training.

## Acknowledgements

We thank the anonymous reviewers and area chair for their constructive feedback. We are also grateful to our colleagues in the NAVER Cloud Hyperscale AI Vision Understanding Team for helpful discussions and support.

## Impact Statement

This paper presents work whose goal is to advance the field of machine learning. There are many potential societal consequences of our work, none of which we feel must be specifically highlighted here.

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

# Appendix

**Table of Contents**

# A. Experimental Details

## A.1. Experimental Setup

This section describes the experimental setup shared across all experiments unless otherwise specified.

### A.1.1. TRAINING DATASETS

**Vision–Language Model Training Data.** For vision–language model experiments, we conduct instruction tuning on Vision-FLAN (Xu et al., 2024b). For experiments at the 3B scale, we use 136 tasks out of the full set of 187 Vision-FLAN tasks for training. All comparison methods share the same dataset splits and training data.

**Language Model Training Data.** For text-only language model experiments, we conduct instruction tuning on the FLAN dataset (Wei et al., 2022). We use a subset of 66 instruction tasks for training in all language model experiments. All baselines and MERIT variants are trained on the same FLAN task mixture.

### A.1.2. EVALUATION BENCHMARKS

**Multimodal Benchmarks.** Our primary evaluation suite, used for every headline comparison in the main text (Sections 5–6), consists of eight widely used multimodal benchmarks that collectively assess general reasoning, open-ended understanding, text-centric perception, and expert-level reasoning: SeedBench (Li et al., 2024a) and MMBench (Liu et al., 2025) for general multimodal reasoning and compositional generalization; LLaVA-Wild (Liu et al., 2023) and MMVet (Yu et al., 2024c) for open-ended and fine-grained multimodal understanding; TextVQA (Singh et al., 2019) for text-centric visual understanding; AI2D (Kembhavi et al., 2016) for diagram-based reasoning; MathVista (Lu et al., 2024) for mathe-

matical reasoning; and MMMU (Yue et al., 2024) for expert-level multimodal reasoning across multiple images. For each benchmark, we report the standard metric defined by the dataset. The clustering-strategy ablation in Appendix C.3 adopts a broader 11-benchmark protocol that additionally includes MME (Fu et al., 2025), HallusionBench (Guan et al., 2024), DocVQA (Mathew et al., 2021), and MIABench (Qian et al., 2025) for robustness to evaluation protocol; LLaVA-Wild is excluded there because it is generation-only and does not fit the MCQA-centric aggregation used in that ablation.

**Text-Only Benchmarks.** For language models, we evaluate on a diverse set of text-only benchmarks covering reasoning, commonsense understanding, code generation, and cross-lingual generalization. These include MMLU (Hendrycks et al., 2021), HellaSwag (Zellers et al., 2019), WinoGrande (Sakaguchi et al., 2020), ARC-C (Clark et al., 2018), HumanEval (Chen et al., 2021), BoolQ (Clark et al., 2019), GPQA (Rein et al., 2024), and XNLI (Conneau et al., 2018). We follow the standard evaluation protocols for all benchmarks.

### A.1.3. BASELINES

**Joint Training.** All datasets within each setting are trained jointly as a single corpus. We evaluate joint training for 0.5, 1, and 2 epochs to cover different optimization regimes. To give the joint baseline the strongest possible footing, we additionally grid-search its per-device batch size and report the best-performing configuration; MERIT branches use a single fixed batch size across all runs. Consequently, any reported gap understates MERIT's advantage under matched tuning budgets.

**Random Split.** Datasets are randomly partitioned into 2, 4, or 8 groups. Each group is trained independently and the resulting checkpoints are merged via weight averaging.

**Uniform Soup.** We include a uniform model soup baseline following Model Soups (Wortsman et al., 2022). Multiple models are trained on the full dataset using different random data orders and merged by uniform averaging. We report soups constructed from 2, 3, and 4 models.

**Conflict-Induced Split.** We consider a conflict-induced split baseline based on greedy maximization of inter-group gradient disagreement. Starting from the most conflicting task pair, remaining tasks are assigned to minimize average cosine similarity within each group. Models are trained independently and merged using the same averaging procedure.

### A.1.4. IMPLEMENTATION DETAILS

**Language Model Experiments** For text-only experiments, we use Qwen2.5-3B (Qwen et al., 2025). Only language model parameters are trained, and all other settings follow the MERIT pipeline.

**Vision–Language Model Experiments** For vision–language experiments, we use Qwen2.5-VL-3B-Instruct (Bai et al., 2025), initialized from a standard two-stage LLaVA-style training recipe. During fine-tuning, the vision encoder and multimodal projector are frozen, and only the language model decoder is updated. Images are processed at a maximum resolution of $784 \times 784$ pixels. Unless otherwise specified, all methods share the same initialization and training configuration.

### A.2. Large-Scale Vision–Language Model Experiments

This section provides details of the two base models and the data mixture used in our large-scale (7B) vision–language model experiments (Section 6.1). Unless stated otherwise, all training procedures follow the setup described in Appendix A.1.

**Overview.** For the 7B-scale study, we do not start from a released vision–language checkpoint; instead, we reuse the vision encoder from Qwen2.5-VL and pair it with the Qwen2.5-7B-Instruct language model, building our own merge-ready vision–language base from this combination via a LLaVA-style training recipe. The vision encoder is kept frozen throughout. The multimodal projector is tuned only while building the (scaled) base VLM; during the subsequent FFT stage (both Joint FFT and MERIT) it is frozen as well, so that this stage updates the language-model parameters alone. We consider two such base checkpoints of increasing strength, and on top of each we perform further full fine-tuning (FFT) on the same large-scale multimodal mixture of 1.6M examples spanning 176 dataset/task sources. Each source (or task-specific subset when a dataset provides multiple tasks) is treated as a dataset unit, following the definition in Section 4, and serves as the

basic unit for gradient conflict estimation and partitioning in MERIT. Both the 1.6M FFT mixture and the training budget are held identical across Joint FFT and MERIT.

**Base VLM and Scaled base VLM.**     The *Base VLM* follows a standard two-stage LLaVA-style recipe: feature alignment followed by a 0.7M-example instruction-tuning stage (*stage 2*). The *Scaled base VLM* additionally inserts a high-quality knowledge-learning stage before instruction tuning, following recent recipes (Li et al., 2024b; 2025): starting from feature alignment, we train on a 2.9M-example re-captioned corpus (*stage 1.5*) and then on the same 0.7M instruction-tuning data (*stage 2*), for 3.6M examples in total. For stage 1.5 we use the publicly available LLaVA-ReCap-CC3M dataset (`https://huggingface.co/datasets/lmms-lab/LLaVA-ReCap-CC3M`). Both checkpoints are then fine-tuned on the same 1.6M FFT mixture described below, so that the only difference between the two builds is the strength of the initialization.

**Data Sources (1.6M FFT mixture).**     The large-scale FFT mixture is drawn from publicly available datasets hosted on Hugging Face:

- `https://huggingface.co/datasets/X2FD/LVIS-Instruct4V`
- `https://huggingface.co/datasets/Vision-Flan/vision-flan`
- `https://huggingface.co/datasets/HuggingFaceM4/the_cauldron`
- `https://huggingface.co/datasets/zhiqings/LLaVA-Human-Preference-10K`
- `https://huggingface.co/datasets/VictorSanh/LrvInstruction`
- `https://huggingface.co/datasets/laion/gpt4v-dataset`
- `https://huggingface.co/datasets/ys-zong/VLGuard`

For Vision-FLAN and The Cauldron we use only a curated subset of the upstream task collection. The exact list of 176 task-unit identifiers and scripts will be released at `https://github.com/naver-ai/merit`.

**Preprocessing and Mixture Construction.**     We convert all datasets into a unified multimodal instruction format compatible with our LLaVA-style training pipeline (image + instruction + response). When a dataset provides multiple task-specific subsets (as in Vision-FLAN and The Cauldron), each subset is preserved as its own task unit, yielding 176 units in total. We apply standard pre-processing and concatenate the units into a single 1.6M-example mixture used for FFT.

**Reproducibility.**     The datasets are publicly available under their original licenses. More details, including the full base-model training configurations, will be released at `https://github.com/naver-ai/merit`. Multi-seed reproducibility of these 7B results is reported in Appendix C.2.

## B. Merge-Readiness Diagnostics

This section consolidates four complementary diagnostics that collectively verify the merge-ready initialization assumption underlying MERIT.

### B.1. Linear Mode Connectivity

We evaluated loss barriers along all pairwise and branch-to-merged interpolation paths in the 3B 2D split ($K=4$) using 21 evenly spaced interpolation points ($\alpha \in [0, 1]$, step 0.05). The barrier is defined as $\max_\alpha[L(\theta(\alpha)) - ((1-\alpha)L(0) + \alpha L(1))]$, measuring the worst-case deviation above the linear interpolation.

### B.2. Weight Perturbation Robustness

We applied isotropic Gaussian perturbations ($\sigma \in \{0.01, 0.05, 0.1\}$) to both the merged and jointly trained models at epochs 1–3. The merged model exhibits consistently lower loss sensitivity and lower flatness AUC (area under the $\Delta L$–$\sigma$ curve) at all perturbation levels and all epochs (Figure 4), confirming that the merged solution sits in a flatter region of the loss landscape.

*Table 5.* Linear Mode Connectivity (2D split, $K{=}4$). All 10 barriers are exactly 0, confirming that branches remain in a shared flat basin with no loss barrier.

| Path | $L(\alpha{=}0)$ | $\min_\alpha L(\alpha)$ | $L(\alpha{=}1)$ | Barrier |
|---|---|---|---|---|
| *Branch $\leftrightarrow$ Branch (pairwise)* | | | | |
| b0 $\leftrightarrow$ b1 | 2.306 | 1.901 ($\alpha{=}0.70$) | 1.993 | 0.0 |
| b0 $\leftrightarrow$ b2 | 2.306 | 2.244 ($\alpha{=}0.35$) | 2.468 | 0.0 |
| b0 $\leftrightarrow$ b3 | 2.306 | 1.959 ($\alpha{=}0.65$) | 2.046 | 0.0 |
| b1 $\leftrightarrow$ b2 | 1.993 | 1.972 ($\alpha{=}0.15$) | 2.468 | 0.0 |
| b1 $\leftrightarrow$ b3 | 1.993 | 1.792 ($\alpha{=}0.45$) | 2.046 | 0.0 |
| b2 $\leftrightarrow$ b3 | 2.468 | 2.040 ($\alpha{=}0.95$) | 2.046 | 0.0 |
| *Branch $\rightarrow$ Merged* | | | | |
| b0 $\rightarrow$ merged | 2.306 | 1.973 ($\alpha{=}1.0$) | 1.973 | 0.0 |
| b1 $\rightarrow$ merged | 1.993 | 1.874 ($\alpha{=}0.50$) | 1.973 | 0.0 |
| b2 $\rightarrow$ merged | 2.468 | 1.973 ($\alpha{=}1.0$) | 1.973 | 0.0 |
| b3 $\rightarrow$ merged | 2.046 | 1.944 ($\alpha{=}0.60$) | 1.973 | 0.0 |

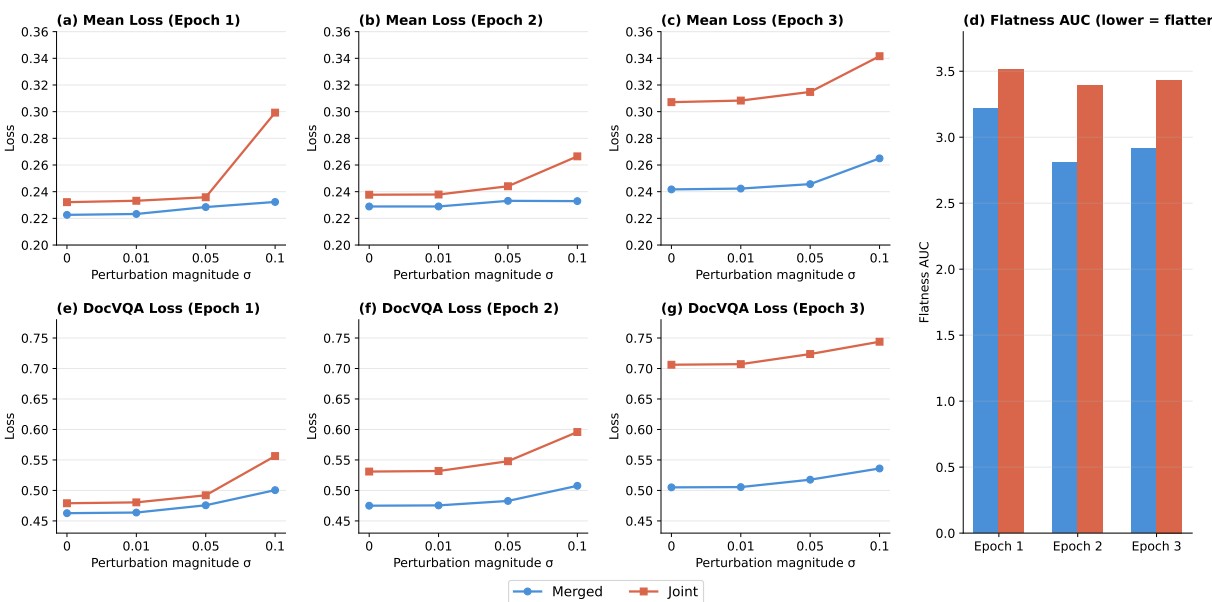

*Figure 4.* Weight perturbation robustness. The merged model (blue) exhibits consistently lower loss sensitivity than the jointly trained model (red) at all perturbation levels and epochs, providing direct evidence of a flatter loss landscape.

### B.3. Displacement Contraction

By convexity of the squared norm, weight averaging contracts the merged model toward the shared initialization: $\|\bar{\theta}_w - \theta^{(0)}\|^2 \leq \sum_i w_i \|\theta_i - \theta^{(0)}\|^2$. Table 6 confirms this empirically: the merged model remains 2–3× closer to $\theta^{(0)}$ throughout training, with the ratio widening monotonically.

### B.4. Training Loss vs. Generalization

The merged model has substantially higher training loss than joint training yet achieves better held-out performance, the classic signature of implicit regularization from merging. The gap widens monotonically over training (Table 7).

## C. Additional Analyses and Ablations

### C.1. Visualization of Dataset-Level Gradients

Figure 5 projects the dataset-level gradients at $\theta^{(0)}$ to two dimensions via t-SNE. Tasks of the same type (e.g., VQA, image classification, and captioning) tend to cluster together, while different types occupy distinct regions of the gradient space.

*Table 6.* Parameter displacement from shared initialization over training epochs.

| Epoch | $\|\theta_{\text{joint}} - \theta^{(0)}\|_2$ | $\|\bar{\theta}_{\text{merge}} - \theta^{(0)}\|_2$ | Ratio |
|---|---|---|---|
| 0.5 | 13.73 | 5.65 | 2.43× |
| 1.0 | 19.73 | 7.50 | 2.63× |
| 2.0 | 28.15 | 10.11 | 2.78× |
| 6.0 | 34.61 | 11.87 | 2.92× |

*Table 7.* Training loss on the full mixture over epochs.

| Epoch | $L_{\text{joint}}$ | $L_{\text{merged}}$ | Gap |
|---|---|---|---|
| 0.5 | 0.709 | 1.198 | +0.489 |
| 1.0 | 0.560 | 1.172 | +0.611 |
| 2.0 | 0.370 | 1.167 | +0.797 |
| 3.0 | 0.205 | 1.202 | +0.998 |
| 6.0 | 0.064 | 1.330 | +1.266 |

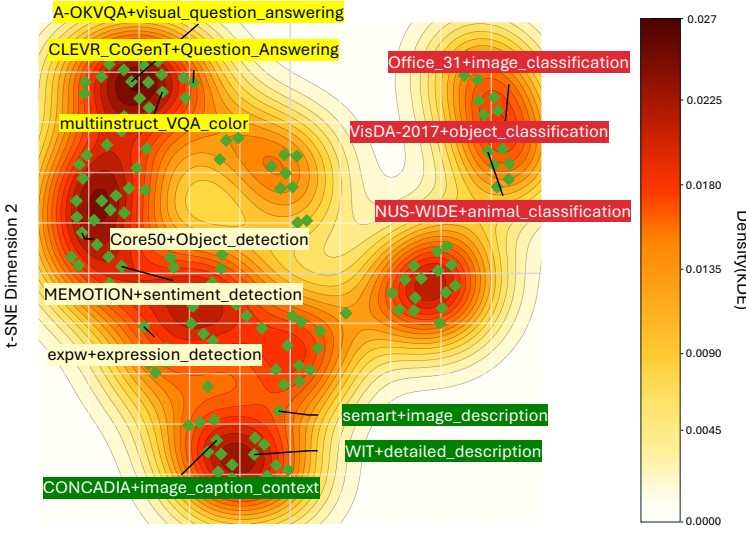

*Figure 5.* Two-dimensional t-distributed stochastic neighbor embedding (t-SNE) (Van der Maaten & Hinton, 2008) of dataset-level gradients at $\theta^{(0)}$ (Qwen2.5-VL-3B, 136 Vision-FLAN tasks), overlaid with kernel density estimation (KDE) contours. Each marker is one task. Labeled examples are colored by task type: VQA (yellow), image classification (red), and description/captioning (green).

This task-type structure is what MERIT's conflict-aware split exploits: gradients that disagree in direction fall into separable regions, so partitioning along the PCA conflict axes groups compatible datasets together.

### C.2. Robustness and Multi-Seed Reproducibility

We assess MERIT's reproducibility under training randomness at both scales: (i) the 3B primary comparison over five independent seeds with a paired Wilcoxon test, and (ii) a 7B replication over three independent seeds. Per-seed comparisons against alternative post-hoc merging operators on the same 7B 2D branches are provided separately in Appendix C.5, as they target a different question (aggregation rule choice rather than training-seed variability).

#### C.2.1. 3B FIVE-SEED COMPARISON

We focus on the primary comparison used throughout the paper: MERIT (CosSim PCA, 3D split, 8 groups) versus Joint training (1 ep). Each method is evaluated over five independent runs with different random seeds, on the same eight benchmarks grouped into four categories (General MCQA, User Preference & Fluency, Text-Rich VQA, Image Reasoning). The reported Avg. denotes the mean score across all eight benchmarks.

**Per-run results.** Table 8 reports per-run results for both methods. MERIT consistently outperforms Joint training on the 8-benchmark average across all five independent runs.

**Statistical Test.** To further quantify the statistical significance of this improvement, we apply a paired Wilcoxon signed-rank test on the per-run averaged scores. Using the five paired observations, the test yields a one-sided $p$-value of 0.03125,

*Table 8.* Per-run results on the 3B setting comparing **MERIT (CosSim PCA, 3D split, 8 groups)** against **Joint training (1 ep)** over five independent runs. MERIT wins on the 8-benchmark average in all five runs (bold).

| Run | Method | General MCQA | | User Preference & Fluency | | Text-Rich VQA | | Image Reasoning | | Avg. |
|-----|--------|:------------:|:-:|:-------------------------:|:-:|:-------------:|:-:|:---------------:|:-:|:----:|
| | | SeedBench | MMBench | LLaVA-W | MMVet | TextVQA | AI2D | MathVista | MMMU | |
| Run 1 | Joint (1 ep) | 68.6 | 80.1 | 41.8 | 38.5 | 68.3 | 62.4 | 35.0 | 40.9 | 54.5 |
| | MERIT-3D | 70.4 | 80.3 | 52.2 | 38.1 | 75.1 | 63.1 | 35.7 | 42.1 | **57.1** |
| Run 2 | Joint (1 ep) | 69.6 | 80.6 | 41.7 | 36.0 | 67.9 | 62.4 | 34.0 | 41.9 | 54.3 |
| | MERIT-3D | 70.6 | 80.1 | 53.7 | 38.8 | 75.0 | 62.1 | 35.9 | 43.3 | **57.4** |
| Run 3 | Joint (1 ep) | 69.1 | 80.4 | 41.5 | 34.9 | 68.0 | 62.2 | 34.2 | 42.1 | 54.1 |
| | MERIT-3D | 70.5 | 79.5 | 49.9 | 36.5 | 75.1 | 62.8 | 34.3 | 42.8 | **56.4** |
| Run 4 | Joint (1 ep) | 69.3 | 80.7 | 42.3 | 36.8 | 67.9 | 63.6 | 33.8 | 42.5 | 54.6 |
| | MERIT-3D | 70.5 | 80.2 | 52.7 | 38.0 | 75.3 | 61.5 | 34.7 | 42.3 | **56.9** |
| Run 5 | Joint (1 ep) | 69.5 | 80.5 | 42.1 | 35.8 | 67.7 | 62.6 | 34.0 | 41.9 | 54.3 |
| | MERIT-3D | 70.3 | 80.5 | 51.7 | 37.1 | 75.4 | 63.2 | 36.6 | 43.2 | **57.3** |

which is statistically significant at the $5\%$ level ($p < 0.05$).

### C.2.2. 7B THREE-SEED REPLICATION

To verify that the gains persist at scale, we repeated the full training pipeline (joint FFT and MERIT-2D) over three independent training seeds on the 7B setting. Table 9 shows that MERIT outperforms joint training in all three runs, with the same gain pattern as the 3B setting (open-ended reasoning benchmarks up, factual benchmarks within noise).

*Table 9.* 7B results over three independent training seeds (Qwen2.5-VL-7B, 176 tasks). MERIT outperforms joint training in all three runs.

| Run | Method | General MCQA | | User Preference & Fluency | | Text-Rich VQA | | Image Reasoning | | Avg. |
|-----|--------|:------------:|:-:|:-------------------------:|:-:|:-------------:|:-:|:---------------:|:-:|:----:|
| | | SeedBench | MMBench | LLaVA-W | MMVet | TextVQA | AI2D | MathVista | MMMU | |
| Run 1 | Joint FFT | 70.1 | 76.2 | 58.8 | 32.5 | 73.8 | 58.6 | 32.9 | 36.4 | 54.9 |
| | MERIT-2D | 70.6 | 75.7 | 59.2 | 35.1 | 72.4 | 58.4 | 35.4 | 36.1 | **55.4** |
| Run 2 | Joint FFT | 70.3 | 75.9 | 60.9 | 34.0 | 74.1 | 58.5 | 31.8 | 36.2 | 55.2 |
| | MERIT-2D | 70.8 | 75.5 | 63.0 | 34.1 | 72.0 | 58.6 | 31.9 | 36.2 | **55.3** |
| Run 3 | Joint FFT | 70.1 | 75.9 | 60.8 | 34.1 | 77.1 | 58.6 | 31.7 | 35.9 | 55.5 |
| | MERIT-2D | 70.4 | 75.3 | 60.1 | 35.6 | 76.3 | 58.8 | 33.0 | 36.4 | **55.7** |

## C.3. Clustering Strategies for Dataset Partitioning

*Table 10.* Ablation study on clustering strategies for dataset partitioning. We compare our PCA-based MERIT (Ours) against K-means clustering with different numbers of clusters.

| Method | MMBench | MME | SeedBench | MathVista | Hallusion | AI2D | MMVet | MIABench | MMMU | TextVQA | DocVQA | Avg. |
|--------|:-------:|:---:|:---------:|:---------:|:---------:|:----:|:-----:|:--------:|:----:|:-------:|:------:|:----:|
| **Ours (MERIT, 1D)** | 80.0 | 1899.4 (82.6) | 71.0 | 36.5 | 54.4 | 62.1 | 35.0 | 34.4 | 41.4 | 72.4 | 50.0 | **56.3** |
| **Ours (MERIT, 2D)** | 78.4 | 1910.3 (83.0) | 70.8 | 36.0 | 53.0 | 61.5 | 36.6 | 37.3 | 40.7 | 74.1 | 50.8 | **56.6** |
| **Ours (MERIT, 3D)** | 80.1 | 1872.8 (81.4) | 70.5 | 35.4 | 54.2 | 62.5 | 37.7 | 39.2 | 42.7 | 75.2 | 50.4 | **57.2** |
| K-means ($k = 2$) | 80.1 | 1834.5 (79.8) | 70.2 | 34.3 | 55.1 | 62.1 | 36.9 | 34.4 | 38.6 | 71.2 | 49.3 | 55.5 |
| K-means ($k = 4$) | 79.7 | 1853.4 (80.6) | 70.9 | 35.2 | 56.2 | 61.9 | 34.4 | 35.4 | 42.3 | 73.7 | 50.2 | 56.4 |
| K-means ($k = 8$) | 79.4 | 1833.1 (79.7) | 70.4 | 35.1 | 53.6 | 62.1 | 35.1 | 39.4 | 41.1 | 75.0 | 50.4 | 56.5 |

Table 10 compares PCA-based MERIT against K-means clustering ($k \in \{2, 4, 8\}$) on the same dataset-level gradient representations, with identical training and merging procedures. MERIT consistently outperforms K-means and scales monotonically with dimensionality, while K-means plateaus at a lower aggregate and exhibits non-monotonic per-benchmark patterns (e.g., MathVista, HallusionBench, MMVet). This confirms that aligning partitions with dominant conflict directions is more effective than generic distance-based clustering.

**Random partitioning as a complementary baseline.** Random partitioning followed by weight merging can already outperform joint training: under the flat-basin setting (Appendix F.1), any non-trivial dispersion among checkpoints yields a deterministic quadratic gain through variance cancellation, even when splits are random. This is consistent with Model Soups (Wortsman et al., 2022), which empirically observe that averaging compatible checkpoints reduces sharpness. However, random partitioning does not control *where* variance is introduced: the displacement covariance $\Sigma_\delta = \frac{1}{K} \sum_i (\delta_i - \bar{\delta})(\delta_i - \bar{\delta})^\top$ is generally unstructured and need not align with dominant curvature directions, limiting the Hessian-weighted gain $\mathcal{G}_{\text{var}}$. MERIT's PCA-based splitting explicitly concentrates dispersion along high-curvature conflict axes, yielding systematically larger gains (Tables 10 and main text Table 2).

### C.4. Calibration Dataset Size Sensitivity

We fix calibration size to 200 samples per dataset throughout the paper. Table 11 reports calibration sensitivity across 100 Vision-FLAN tasks; we restrict this analysis to the 100 tasks that contain at least 1,000 samples, since the reference gradient requires $n=1,000$. At $n=200$, the mean cosine similarity with the full 1000-sample reference gradient is $0.847$ (std $0.106$), with diminishing returns beyond this point.

*Table 11.* Gradient calibration sensitivity across 100 Vision-FLAN tasks. **Left**: cosine similarity between the calibration-subset gradient and the full 1000-sample reference, averaged over 100 tasks. **Right**: the same values as a function of $n$, with $n=200$ (used) marked.

| $n$ | Mean cos | Std |
|---|---|---|
| 50 | 0.683 | 0.176 |
| 100 | 0.762 | 0.147 |
| 150 | 0.814 | 0.121 |
| 200 | 0.847 | 0.106 |
| 250 | 0.871 | 0.095 |
| 300 | 0.891 | 0.087 |
| 400 | 0.920 | 0.076 |
| 500 | 0.940 | 0.070 |
| 700 | 0.968 | 0.062 |
| 1000 | 1.000 | 0.000 |

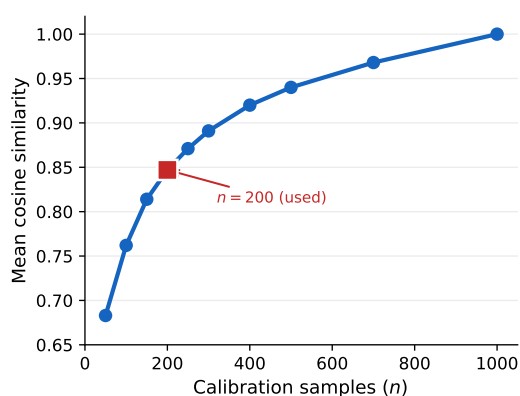

### C.5. Post-Hoc Merging Baselines

This section provides per-seed results for the post-hoc merging comparison summarized in Section 6.4. Each baseline is applied as a drop-in replacement for token-weighted averaging on MERIT's 7B 2D branches ($K=4$) over three independent seeds, using default hyperparameters from each method's official repository.

*Table 12.* Per-seed results for post-hoc merging baselines on MERIT's 7B 2D branches. The Mean column matches Figure 3 in the main text.

| Merging method | Seed 1 | Seed 2 | Seed 3 | Mean |
|---|---|---|---|---|
| Token-Weighted Avg | **55.4** | **55.3** | **55.7** | **55.5** |
| TIES (Yadav et al., 2023) | 48.8 | 50.0 | 52.4 | 50.4 |
| STAR (Lee et al., 2025b) | 38.4 | 42.5 | 42.2 | 41.0 |
| TSV (Gargiulo et al., 2025) | 45.5 | 46.2 | 46.5 | 46.1 |
| Iso-CTS (Marczak et al., 2025) | 42.9 | 45.1 | 48.3 | 45.4 |

## D. Additional Qualitative Analysis

### D.1. Short-Answer Collapse on LLaVA-Wild

Joint training on the 3B setting shows a large LLaVA-Wild degradation ($53.2 \rightarrow 41.9$–$42.8$) due to short-answer collapse: the model produces overly terse responses under heterogeneous mixtures (Dai et al., 2023; Xu et al., 2024b). The same effect appears at 7B scale, where Joint FFT on the Scaled base collapses LLaVA-Wild from $67.1$ to $50.2$, whereas MERIT

*Table 13.* Additional qualitative examples from LLaVA-Wild.

| Input (Image + Question) | Model Responses |
| --- | --- |
| 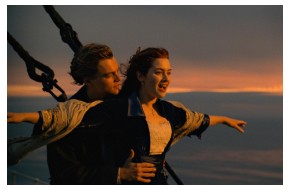 **Q:** What is the significance of this scene in the context of the movie? | **Joint:** love 

 **MERIT (1D):** This scene is significant because it is the first time the couple is seen together. 

 **MERIT (2D):** The scene captures a romantic moment between the two characters as they look out at the ocean together, with the sunset adding emotional tone. 

 **MERIT (3D):** The scene depicts a key romantic moment between Jack and Rose, emphasizing their emotional connection and the beginning of their journey together, reinforced by the sunset setting. |
| 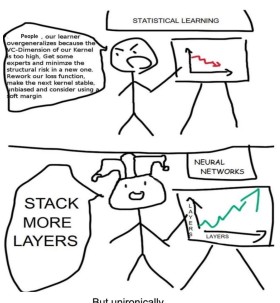 **Q:** Can you explain why this is funny. Think about it step-by-step. | **Joint:** The joke is that the person is trying to convey that the VC is trying to get more money for the company, but the company is not stable enough to do so. 

 **MERIT (1D):** The image is funny because it is a cartoon. 

 **MERIT (2D):** The funny part of this is that the cartoon character is trying to explain something to the other character, but the other character is not listening. 

 **MERIT (3D):** The funny part of this is that the cartoon character is talking about a computer science concept, but he is using a very simple and basic example. The character is talking about stacking layers, but he is only stacking two layers. This is a very basic example of a computer science concept, and it is not very interesting or complex. The fact that the character is using such a simple example makes it funny because it is not what one would expect to see in a computer science context. |
| 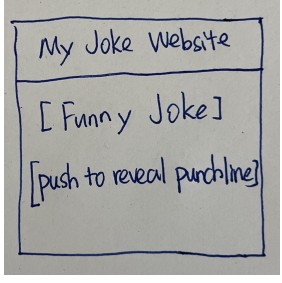 **Q:** Describe this sketch in detail. | **Joint:** This is a sketch of a website. 

 **MERIT (1D):** The sketch is a white board with a blue line and text. The text is "My Joke Website" and "Funny Joke" and "Push to reveal punchline". 

 **MERIT (2D):** A white board with a blue line and text. 

 **MERIT (3D):** A white board with a blue line and the words "My Joke Website" written on it. |

preserves the base model's open-ended quality (66.2). In both cases, MERIT mitigates the collapse by separating conflicting subsets before fine-tuning, yielding more stable open-ended behavior; representative examples are shown in Table 13.

### D.2. Qualitative Examples on Multimodal Reasoning

The examples above show that joint-trained models tend to produce terse or generic responses, while MERIT models provide more informative and context-aware answers. Higher decomposition dimensionality often (though not uniformly) yields richer and more context-aware answers, with the largest gains on responses that require multi-step reasoning or named-entity recall (e.g., Examples 1 and 2).

## E. Detailed Efficiency Analysis

Table 15 summarizes wall-clock times. Preprocessing is dominated by gradient extraction ($\sim$1.6h for 136 datasets on V100, $\sim$2.0h for 176 on A100); cosine similarity computation adds $\sim$28–38 min and PCA takes $< 1$s. Table 16 confirms that uniform gradient sampling preserves cosine similarity structure with high fidelity ($\rho > 0.98$).

*Table 14.* Hardware and system configuration used for efficiency analysis.

|  | V100 System | A100 System |
|---|---|---|
| **CPU** | | |
| Model | Intel Xeon Gold 5120 @ 2.20GHz | Intel Xeon Gold 6338 @ 2.00GHz |
| Sockets | 2 | 2 |
| Cores | 28 (14 per socket) | 64 (32 per socket) |
| Threads | 56 | 128 |
| **GPU** | | |
| Model | Tesla V100-SXM2-32GB | NVIDIA A100-SXM4-80GB |
| Number of GPUs | 8 | 8 |
| Memory per GPU | 32GB | 80GB |
| Total GPU Memory | 256GB | 640GB |

*Table 15.* Wall-clock time breakdown of MERIT and joint training. Experiments are conducted with a 3B model on the V100 system (136 datasets, 3D split, $K=8$) and a 7B model on the A100 system (176 datasets, 3D split, $K=8$, measured with sequential branch execution). The 7B robustness results in Table 9 use the 2D split ($K=4$) configuration.

| Stage | V100 System | A100 System |
|---|---|---|
| Basin preparation (pre-alignment) | 4h 24m / 20h 13m (LLaVA recipe stage 1/2) | 39 h (Base VLM 7B) |
| Gradient extraction | 1h 38m (136 datasets) | 1h 59m (176 datasets) |
| Cosine similarity matrix computation | 28m | 38m |
| PCA on cosine similarity matrix | 0.7s (load) + 0.2s (compute) | 0.6s (load) + 0.12s (compute) |
| Joint training (1 ep) | 4h 22m | 43 h 18m |
| Joint training (2 ep) | 8h 40m | – |
| MERIT (CosSim PCA, 3D split) | 5h 24m | 43 h 39m |

The similarity matrix is reusable: adding $m$ new datasets to an existing $T$-dataset collection requires only $O(Tm)$ cross-similarity computations plus a negligible PCA update, avoiding full $O((T+m)^2)$ recomputation.

Counting branch training and merging (preprocessing reported separately above), MERIT-3D completes in 5h24m on V100, faster than 2-epoch joint training (8h40m) and only modestly above 1-epoch joint training (4h22m). At 7B scale on A100, MERIT adds just 21 minutes ($0.8\%$) over 1-epoch joint training (43h39m vs. 43h18m). This one-time, amortizable preprocessing cost removes the need for synchronous gradient communication, making the approach well suited to decentralized or bandwidth-constrained environments.

### E.1. Comparison with Centralized Gradient Methods

We provide a quantitative feasibility analysis for centralized per-step gradient conflict resolution methods at our experimental scale ($T=136$ tasks, 3B parameters).

**PCGrad** (Yu et al., 2020) requires $T$ backward passes + $\binom{T}{2}$ = 9,180 pairwise projections per step. Storing 136 full gradients requires $\sim$816 GB (fp16), exceeding both our V100$\times$8 (256 GB) and A100$\times$8 (640 GB) systems. A conservative wall-clock estimate is >17 days per epoch. **GradNorm** (Chen et al., 2018) requires all $T=136$ tasks to contribute individual loss terms at every step with centralized coordination. Both methods were originally evaluated on 2–10 tasks on models orders of magnitude smaller, and to our knowledge neither has been applied to >100 tasks on billion-parameter models.

## F. Theoretical Analysis

This appendix collects the formal assumptions, proofs, and extended analyses behind the three implications stated in Section 3.

### F.1. Quadratic Analysis of Merging in Flat PCA-Structured Basins

This section makes precise the variance-reduction and spectral-filtering implications. We analyze the effect of merging through a deterministic quadratic gain from variance reduction, controlled remainder and dataset-split mismatch terms, and an optimization and regularization benefit over joint training.

*Table 16.* Similarity between cosine similarity matrices computed using full gradients and uniformly sampled gradients (stride $s = 5$).

| Metric | Value |
|---|---|
| Pearson correlation ($r$) | 0.9884 |
| Spearman rank correlation ($\rho$) | 0.9882 |
| Mean absolute difference | 0.0216 |
| Root mean squared error (RMSE) | 0.0355 |

*Table 17.* Computational comparison of conflict-resolution methods at $T{=}136$ tasks on a 3B model.

| Method | When resolved | Per-step overhead | Communication |
|---|---|---|---|
| PCGrad | Per-step | $O(T^2){\times}d$; $\sim$816 GB | All-reduce + $T$ backpasses |
| GradNorm | Per-step | $O(T)$ forward | All-reduce + per-task loss |
| MERIT | One-time ($\sim$2h) | Zero | Zero (one-shot merge) |

**Setting and notation.** Let $\theta \in \mathbb{R}^d$ be the model parameters and $L : \mathbb{R}^d \to \mathbb{R}$ the true population loss. We consider $K \geq 2$ checkpoints $\theta_1, \dots, \theta_K$ obtained by fine-tuning a common pretrained initialization $\theta^{(0)}$ on different dataset splits via a two-phase procedure. In Phase I, pretraining and instruction tuning yield the merge-ready $\theta^{(0)}$; in Phase II, the $K$ branches are fine-tuned independently on their splits.

Because of the shared Phase I training, all checkpoints lie in one loss basin, a property widely observed in pretrained models and captured by linear mode connectivity (Garipov et al., 2018; Draxler et al., 2018; Qin et al., 2022). We fix a reference point $\theta^\star$ inside this basin, a near-stationary point that serves as the center of the local analysis. For clarity we assume

$$\nabla L(\theta^\star) = 0,$$

noting that approximate stationarity $\|\nabla L(\theta^\star)\| \leq \varepsilon$ only adds $O\big(\varepsilon \max_i \|\delta_i\|\big)$ terms that do not change the conclusions.

We write the deviations from $\theta^\star$ and the parameter average as

$$\delta_i := \theta_i - \theta^\star, \quad i = 1, \dots, K, \qquad \bar{\theta} := \frac{1}{K} \sum_{i=1}^{K} \theta_i, \qquad \bar{\delta} := \bar{\theta} - \theta^\star,$$

and denote the Hessian of the true loss at $\theta^\star$ by $H := \nabla^2 L(\theta^\star)$. We present the uniform-weight case ($w_i = 1/K$); the token-weighted case used in the algorithm is identical after replacing $\frac{1}{K} \sum_i$ with $\sum_i w_i$.

### F.1.1. TWO-PHASE BASIN CONFINEMENT AND LOCAL QUADRATICITY

The analysis rests on a local quadratic approximation of the loss, which is naturally induced by pretrained initialization and two-phase training.

**Assumption 1 (Flat basin and local quadratic model).** There exists a convex neighborhood $\mathcal{B}$ of $\theta^\star$ such that:

(a) (*Basin confinement*) All checkpoints and their average lie in $\mathcal{B}$, i.e., $\theta_i \in \mathcal{B}$ for all $i$ and $\bar{\theta} \in \mathcal{B}$, as expected from a shared merge-ready initialization and confirmed empirically throughout training (Appendix B).

(b) (*Flatness*) For all $\theta \in \mathcal{B}$, $0 \preceq \nabla^2 L(\theta) \preceq \lambda_{\max} I_d$ for some $\lambda_{\max} > 0$, reflecting the flattening effect of pretraining on downstream landscapes (Neyshabur et al., 2020).

(c) (*Controlled remainder*) The loss admits a second-order Taylor expansion around $\theta^\star$,

$$L(\theta) = L(\theta^\star) + \tfrac{1}{2}(\theta - \theta^\star)^\top H (\theta - \theta^\star) + R(\theta),$$

with $|R(\theta)| \leq \frac{\rho}{6} \|\theta - \theta^\star\|^3$ for some third-order smoothness constant $\rho > 0$ (Nesterov & Polyak, 2006; Jin et al., 2017; Antonakopoulos et al., 2022).

We write the quadratic surrogate of the true loss as

$$L_Q(\theta) := L(\theta^\star) + \tfrac{1}{2}(\theta - \theta^\star)^\top H (\theta - \theta^\star).$$

STEP 1: QUADRATIC AVERAGING YIELDS A DETERMINISTIC GAIN

Under the quadratic model, averaging never hurts, and the benefit is exactly the spread of the checkpoints measured in the curvature ($H$) geometry.

**Theorem F.1** (Quadratic averaging bound). *Under Assumption 1 and $H \succeq 0$,*

$$L_Q(\bar{\theta}) \ \leq \ \frac{1}{K} \sum_{i=1}^{K} L_Q(\theta_i),$$

*with equality iff all displacements $\delta_i - \delta_j$ lie in $\ker(H)$ (in particular, equality iff $\theta_1 = \cdots = \theta_K$ when $H \succ 0$). The improvement is strict whenever some pair $(i,j)$ has $(\delta_i - \delta_j)$ with a non-zero projection onto $\mathrm{range}(H)$ (the span of eigenvectors with positive eigenvalues), and admits the explicit form*

$$\frac{1}{K} \sum_{i=1}^{K} L_Q(\theta_i) - L_Q(\bar{\theta}) = \underbrace{\frac{1}{4K^2} \sum_{i=1}^{K} \sum_{j=1}^{K} (\delta_i - \delta_j)^\top H (\delta_i - \delta_j)}_{=: \mathcal{G}_{\mathrm{var}} \geq 0}.$$

The term $\mathcal{G}_{\mathrm{var}}$ is a Hessian-weighted variance: merging performs an explicit variance reduction, and the saved loss is larger along sharper (higher-curvature) directions (Izmailov et al., 2018). This is why *where* the checkpoints disagree matters more than *how much*.

STEP 2: ACCOUNTING FOR DATASET-SPLIT MISMATCH

The gain above is stated for the empirical loss seen during training. To extend it to the true objective we write $L(\theta) = \bar{L}(\theta) + \varepsilon(\theta)$, where $\bar{L}$ is the loss averaged over the dataset splits and $\varepsilon$ is the population-level mismatch. A naive Lipschitz bound on the mismatch scales *linearly* ($O(\max_i \|\delta_i - \bar{\delta}\|)$) and could in principle overwhelm the *quadratic* merging gain as checkpoints converge. We avoid this by treating the mismatch as smooth, which makes its penalty quadratic as well.

**Assumption 2 (Second-order smoothness of mismatch).** The mismatch $\varepsilon(\theta)$ is twice continuously differentiable in $\mathcal{B}$; write $H_\varepsilon := \nabla^2 \varepsilon(\bar{\theta})$.

With $\Delta_{\mathrm{split}} := \varepsilon(\bar{\theta}) - \frac{1}{K} \sum_{i=1}^{K} \varepsilon(\theta_i)$, a second-order expansion around $\bar{\theta}$ (using $\sum_i (\theta_i - \bar{\theta}) = 0$) gives

$$\frac{1}{K} \sum_{i=1}^{K} \varepsilon(\theta_i) \approx \varepsilon(\bar{\theta}) + \underbrace{\nabla \varepsilon(\bar{\theta})^\top \Big( \frac{1}{K} \sum_i (\theta_i - \bar{\theta}) \Big)}_{=0} + \frac{1}{2K} \sum_{i=1}^{K} (\theta_i - \bar{\theta})^\top H_\varepsilon (\theta_i - \bar{\theta}),$$

so the penalty is bounded quadratically,

$$|\Delta_{\mathrm{split}}| \ \lesssim \ \frac{1}{2} \lambda_{\max}(H_\varepsilon) \cdot \frac{1}{K} \sum_{i=1}^{K} \|\delta_i - \bar{\delta}\|^2.$$

Hence both the gain $\mathcal{G}_{\mathrm{var}}$ and the penalty $\Delta_{\mathrm{split}}$ are $O\big( \max_i \|\delta_i - \bar{\delta}\|^2 \big)$, and the net effect of merging is decided by the relative spectra of $H$ and $H_\varepsilon$. Our PCA-structured split (Step 4) deliberately places dispersion in the curvature-bearing subspace of $H$, whereas the mismatch term is residual split-specific disagreement that need not align with those directions, so in this regime the gain dominates the penalty.

STEP 3: EXTENSION TO THE TRUE POPULATION LOSS

**Theorem F.2** (Merging in a flat basin (informal)). *Under Assumptions 1–2 and the second-order mismatch approximation of Step 2, the merged model satisfies*

$$L(\bar{\theta}) \ \leq \ \underbrace{\frac{1}{K} \sum_{i=1}^{K} L(\theta_i)}_{\textit{Expected Individual Loss}} \ - \ \underbrace{\mathcal{G}_{\mathrm{var}}}_{\textit{Quadratic Gain}} \ + \ |\Delta_{\mathrm{HO}}| \ + \ |\Delta_{\mathrm{split}}|, \qquad |\Delta_{\mathrm{HO}}| \leq \frac{\rho}{6} \Big( \|\bar{\delta}\|^3 + \frac{1}{K} \sum_{i=1}^{K} \|\delta_i\|^3 \Big).$$

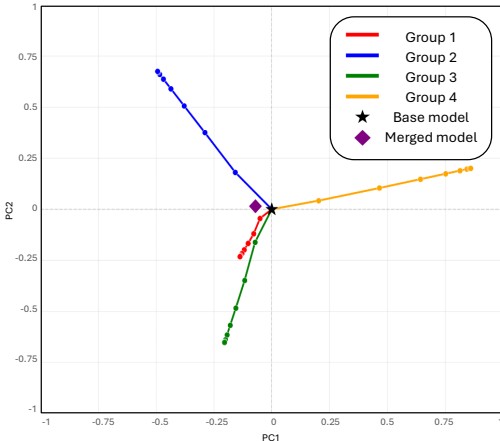

*Figure 6.* PCA-based visualization of dataset groups in gradient space. The split groups extend along distinct, often conflicting directions from the shared initialization, while the merged model lies near the center, reflecting aggregation of complementary updates.

*The gain $\mathcal{G}_{\mathrm{var}}$ is exact (Theorem F.1); the two remainders are the cubic Taylor residual $|\Delta_{\mathrm{HO}}|$ and the second-order (hence approximate) split-mismatch penalty $|\Delta_{\mathrm{split}}|$.*

Merging therefore helps whenever the true-loss curvature along the displacement direction outweighs the mismatch curvature. For small displacements confined to the basin the cubic term is negligible, and since $\mathcal{G}_{\mathrm{var}}$ is maximized by our PCA alignment while $|\Delta_{\mathrm{split}}|$ stays bounded by the smaller curvature of task disagreement, the quadratic gain offsets both remainders.

STEP 4: PCA-STRUCTURED PARTITIONING

We now link the algorithmic choice of cosine-similarity PCA to the amplification of $\mathcal{G}_{\mathrm{var}}$.

**Assumption 3 (PCA-aligned subspace).** Let $U \in \mathbb{R}^{d \times r}$ be an orthonormal basis of the dominant curvature directions, so $H \approx U \Lambda U^{\top} + H_{\perp}$ with $\Lambda \succeq 0$ (Marczak et al., 2025). Decompose $\delta_i = U a_i + b_i$, and assume the split distributes checkpoints so that the mean projection onto the principal subspace is negligible, $\|\bar{a}\| \ll \sqrt{\frac{1}{K} \sum_i \|a_i\|^2}$.

This assumption is justified by gradient accumulation. Under a first-order approximation of fine-tuning with a constant learning rate (here $S$ denotes the number of SGD steps),

$$\delta_i = -\eta \sum_{t=1}^{S} g_{i,t} + r_i, \tag{3}$$

where $r_i$ collects higher-order effects (curvature variation, stochastic noise). Hence, up to second order, $\mathrm{Cov}(\delta_i) \approx \eta^2 \mathrm{Cov}\left(\sum_t g_{i,t}\right)$. Our algorithm runs PCA on the cosine-similarity matrix $C$ (Appendix F.2) and splits along its principal axes, which enlarges the dispersion of $\delta_i$ along the dominant curvature directions $U$ and thus enlarges $\mathcal{G}_{\mathrm{var}}$ relative to a random balanced split. We do not require the gradient-defined subspace to equal the top eigenspace of $H$; it suffices that the two are strongly coupled. Appendix F.4 makes this exact in a tractable linear-quadratic model where the two coincide: there, PCA-aligned splitting maximizes $\mathcal{G}_{\mathrm{var}}$ among all balanced partitions in the $T{=}4$, $d{=}2$ instance, and for general $T$ it provably dominates random partitioning in expectation under a spectral-concentration condition.

**Proposition F.3** (Effect of PCA-structured splitting). *Under Assumption 3, PCA-structured splitting aligns the displacements $\delta_i$ primarily with the high-curvature subspace $U$, so $\mathcal{G}_{\mathrm{var}}$ concentrates along the largest-eigenvalue directions $\Lambda$ and is maximized. In the ideal symmetric case ($\sum_i a_i = 0$), the high-curvature loss component is fully cancelled in the merged model, leaving only the residual loss on the flat subspace $H_{\perp}$.*

STEP 5: OPTIMIZATION ADVANTAGE OVER JOINT TRAINING (CONCEPTUAL)

Beyond a lower loss, merging also eases the *optimization difficulty* of joint training. We give this as an interpretation of the deterministic averaging step rather than a formal guarantee; the SGD experiment in Step 6 provides the matching empirical support.

*Table 18.* Average performance under an SGD learning-rate sweep. MERIT (1D) stays stable across all $\eta$ and MERIT (1D/2D) for $\eta \geq 8\times10^{-5}$, whereas insufficient-epoch joint training collapses at small $\eta$; MERIT (3D) also degrades there, faster than 1D/2D, due to vanishing displacement variance (Appendix F.1, Step 6). Bold denotes the best result per column.

| Method | Learning rate $\eta$ | | | | |
|---|---|---|---|---|---|
| | $3\times10^{-5}$ | $4\times10^{-5}$ | $8\times10^{-5}$ | $2\times10^{-4}$ | $4\times10^{-4}$ |
| Joint (1 ep) | 17.7 | 30.8 | 51.7 | **53.4** | 53.3 |
| Joint (2 ep) | **48.5** | **51.7** | **53.1** | 53.1 | 53.3 |
| MERIT (1D) | 47.6 | 51.3 | 52.9 | 53.1 | 53.7 |
| MERIT (2D) | 18.2 | 32.3 | 51.5 | 53.2 | **53.8** |
| MERIT (3D) | 17.5 | 17.3 | 34.0 | 52.6 | 53.7 |

Joint training is slow for the following reason. Near $\theta^\star$, descent on the full mixture evolves the error $e_t := \theta_t - \theta^\star$ as $e_{t+1} \approx (I - \eta H)\, e_t$, so stability along the sharpest direction caps the learning rate at $\eta < 2/\lambda_{\max}$. When conflicting datasets keep injecting gradient fluctuations along these sharp directions, joint training must keep $\eta$ small to avoid oscillation, which in turn slows progress along the flat directions ($\lambda_{\min} \ll \lambda_{\max}$).

Merging sidesteps this. PCA-structured splitting followed by averaging makes the merged displacement nearly orthogonal to the sharp subspace, $U^\top \bar{\delta} \approx 0$ (Steps 1–4), so high-curvature error is suppressed in one shot instead of being slowly contracted. Conceptually, the remaining optimization is governed by an effective condition number

$$\kappa_{\mathrm{eff}} \approx \frac{\lambda_{\perp,\max}}{\lambda_{\min}} \ll \kappa = \frac{\lambda_{\max}}{\lambda_{\min}},$$

where $\lambda_{\perp,\max}$ is the largest eigenvalue of $H_\perp$. Merging thus trades the slow, stability-limited descent of joint training for an immediate cancellation of the stiffest conflicts.

### STEP 6: EMPIRICAL VERIFICATION UNDER SGD

The displacement model (3) matches SGD exactly, where the update is proportional to accumulated gradients. Adaptive optimizers such as Adam rescale per coordinate via $\hat{m}_t/(\sqrt{\hat{v}_t} + \epsilon)$ and distort this covariance, so the theory of Steps 1–5 is tightest under SGD. We therefore use an SGD learning-rate sweep (Table 18) as a controlled *stress-test of the mechanism*; our main results use AdamW, so this study isolates the variance-reduction effect rather than reproducing those numbers. Two qualitative predictions follow from Steps 4–5 and Proposition F.3.

First, merging should help in the practical regime. For $\eta \in [8\times10^{-5},\, 4\times10^{-4}]$, displacements are large enough that PCA-structured splitting separates datasets along directions of maximal gradient divergence, generating non-trivial inter-group variance that averaging then cancels along high-curvature directions. Table 18 confirms this for 1D and 2D splits.

Second, finer splits should degrade at small $\eta$. As $\eta \to 0$, $\mathcal{G}_{\mathrm{var}} \propto \eta^2 \to 0$ for any partition; in addition, 3D splits give each branch a more homogeneous subset, shrinking $\|\bar{g}_1 - \bar{g}_2\|$ independently of $\eta$. Together these push $\mathcal{G}_{\mathrm{var}} \propto \eta^2 \|\bar{g}_1 - \bar{g}_2\|_H^2$ to zero faster than for 1D/2D. Table 18 shows MERIT (3D) collapsing at $\eta \leq 4\times10^{-5}$, as predicted.

These results match the two predictions above: PCA-aligned splitting concentrates inter-group variance along high-curvature directions (Proposition F.3), and the conditioning benefit of merging (Step 5) keeps MERIT (1D) stable across all $\eta$ and MERIT (1D/2D) stable in the practical regime ($\eta \geq 8 \times 10^{-5}$), even where insufficient-epoch joint training collapses.

### F.2. Theoretical Connection Between Gradient PCA and Cosine-Similarity PCA

This section establishes when PCA on cosine-similarity conflict matrices recovers the same dominant interaction structure as PCA on raw gradients, justifying cosine-similarity PCA as a scale-invariant proxy for the gradient-space analysis above.

#### F.2.1. PROBLEM SETUP AND NOTATION

Let $n$ be the number of datasets and $d$ the parameter dimension. For dataset $i$, let $g_i \in \mathbb{R}^d$ be the population gradient, collected as $G := [g_1, \ldots, g_n]^\top \in \mathbb{R}^{n \times d}$. We compare two constructions:

- **Raw gradient PCA**, equivalent to the eigendecomposition of the Gram matrix (Schölkopf et al., 1998) $K := GG^\top \in \mathbb{R}^{n \times n}$;

*Table 19.* Statistics of dataset-level gradient norms across 136 Vision-FLAN datasets. We report the mean ($\mu$), standard deviation ($\sigma$), coefficient of variation ($\sigma/\mu$), and max-to-min ratio under raw and trimmed settings to assess gradient norm concentration.

| Metric | Raw | 5% Trimmed | 10% Trimmed |
|---|---|---|---|
| Number of datasets | 136 | 124 | 110 |
| Mean gradient norm ($\mu$) | 23.87 | 22.90 | 22.67 |
| Standard deviation ($\sigma$) | 9.27 | 5.27 | 3.88 |
| Coefficient of variation ($\sigma/\mu$) | 0.39 | 0.23 | **0.17** |
| Max / Min ratio | 7.12 | 3.12 | 2.05 |

- **Cosine-similarity PCA**, the eigendecomposition of $C \in \mathbb{R}^{n \times n}$ with $C_{ij} := g_i^\top g_j / (\|g_i\| \|g_j\|)$.

Our goal is to characterize when PCA on $C$ recovers the same dominant subspace as PCA on $G$.

### F.2.2. COSINE SIMILARITY AS A NORMALIZED GRAM MATRIX

**Lemma 1 (Normalization).** With $D := \mathrm{diag}(\|g_1\|, \ldots, \|g_n\|)$, the cosine-similarity matrix satisfies

$$C = D^{-1} K D^{-1}. \tag{4}$$

**Proof.** Immediate from $C_{ij} = K_{ij} / \sqrt{K_{ii} K_{jj}}$. ∎

Cosine similarity is thus a two-sided rescaling of the Gram matrix: it strips out per-dataset gradient magnitude while keeping directional alignment.

### F.2.3. SUBSPACE EQUIVALENCE UNDER GRADIENT NORM CONCENTRATION

Equivalence of the leading eigenspaces of $K$ and $C$ requires the gradient magnitudes to be roughly balanced, a mild condition in large models, where dataset-level gradients are averaged over many samples.

**Assumption 4 (Gradient norm concentration).** There exists $\epsilon \ll 1$ with $\|g_i\| = \bar{\alpha}(1 + \delta_i)$, $|\delta_i| \le \epsilon$, where $\bar{\alpha} > 0$ is a reference scale (e.g., the mean norm).

**Empirical justification.** Across 136 Vision-FLAN datasets (Table 19), raw norms vary, but removing a few extreme datasets sharply improves concentration: the coefficient of variation drops from 0.39 (raw) to 0.23 (5% trimmed) and 0.17 (10% trimmed), with the trimmed max-to-min ratio near 2–3. This supports the small-perturbation regime of Theorem F.4.

**Theorem F.4** (Leading eigenspace equivalence). *Let $V_r(K)$ and $V_r(C)$ be the spans of the top-$r$ eigenvectors of $K$ and $C$. Under Assumption 4, if $K$ has a spectral gap $\lambda_r - \lambda_{r+1} \ge \gamma > 0$, then*

$$\mathrm{dist}\big(V_r(K), V_r(C)\big) = O\left(\frac{\epsilon \|K\|_2}{\gamma}\right), \tag{5}$$

*where $\mathrm{dist}(\cdot, \cdot)$ is the canonical subspace distance via principal angles.*

**Proof Sketch.** Write $D = \bar{\alpha}(I + E)$ with $\|E\|_2 = O(\epsilon)$. Then

$$C = D^{-1} K D^{-1} = \bar{\alpha}^{-2}(I + E)^{-1} K (I + E)^{-1} = \bar{\alpha}^{-2}(K + \Delta), \tag{6}$$

with $\|\Delta\|_2 = O(\epsilon \|K\|_2)$. Scalar multiplication leaves eigenvectors unchanged, so the perturbation is governed by $\Delta$; the Davis–Kahan $\sin \Theta$ theorem (Davis & Kahan, 1970) gives the bound. ∎

### F.2.4. DIRECTIONAL INTERPRETATION VIA NORMALIZED GRADIENTS

With normalized gradients $\tilde{g}_i := g_i / \|g_i\|$ and $\tilde{G} := [\tilde{g}_1, \ldots, \tilde{g}_n]^\top$, we have $C = \tilde{G} \tilde{G}^\top$. Thus cosine-similarity PCA is PCA on unit-sphere-projected gradients, consistent with Theorem F.4: magnitude normalization does not move the leading subspace under norm concentration.

*Table 20.* Direct comparison between MERIT instantiated with raw-gradient PCA and cosine-similarity PCA.

| Method | SeedBench | MMBench | LLaVA-W | MMVet | TextVQA | AI2D | MathVista | MMMU | Avg. |
|---|---|---|---|---|---|---|---|---|---|
| Raw PCA, 1D | 70.9 | 79.9 | 41.5 | 35.0 | 72.5 | **62.7** | 36.2 | 41.3 | 55.0 |
| Raw PCA, 2D | **71.0** | **81.4** | 44.8 | 35.4 | 73.4 | 62.6 | 36.4 | 42.5 | 55.9 |
| Raw PCA, 3D | 70.6 | 81.2 | 40.0 | 35.4 | **75.4** | 62.2 | 33.6 | 41.0 | 54.9 |
| CosSim PCA, 1D | **71.0** | 80.0 | 43.1 | 35.0 | 72.4 | 62.1 | **36.5** | 41.4 | 55.2 |
| CosSim PCA, 2D | 70.8 | 78.4 | 47.4 | 36.6 | 74.1 | 61.5 | 36.0 | 40.7 | 55.7 |
| CosSim PCA, 3D | 70.5 | 80.1 | **52.0** | **37.7** | 75.2 | 62.5 | 35.4 | **42.7** | **57.0** |

*Table 21.* Empirical Hessian–PCA alignment between top-3 PCA conflict directions and top-10 Hessian eigenvectors at $\theta^{(0)}$. Alignment is partial in absolute terms (leading principal angles $55$–$70°$) but well above the Gaussian random baseline ($z \gg 1$) at both scales.

| Metric | Qwen2.5-VL-3B ($d \approx 3$B) | SmolVLM-256M ($d \approx 256$M) |
|---|---|---|
| Raw max $\|\cos\|$ | 0.30 | 0.48 |
| Raw mean top-1 $\|\cos\|$ | 0.136 | 0.332 ($2.4\times\uparrow$) |
| Raw leading principal angle | $69.5°$ | $55.5°$ |
| Raw $z$-score (vs. Gaussian) | 15,386 | 17,973 |
| Centered $z$-score | 19,385 | 15,858 |

### F.2.5. EMPIRICAL AND PRACTICAL IMPLICATIONS

Empirically (Table 20), at 1D and 2D the two PCA constructions give nearly identical averages (within 0.2 points) and the same ordering (2D > 1D), consistent with the leading-subspace equivalence of Theorem F.4; the upward trend on text-heavy benchmarks (e.g., TextVQA, MMVet) appears in both. At 3D the two diverge: cosine-similarity PCA continues to improve while raw-gradient PCA does not, which motivates the scale-invariant cosine construction adopted below. Practically, cosine-similarity PCA is the convenient, scale-invariant choice: it emphasizes directional disagreement and is robust to dataset-specific gradient-norm differences, without altering the dominant interaction subspace. We therefore use it as the default instantiation of MERIT.

### F.3. Empirical Hessian–PCA Alignment

The link between gradient-PCA and Hessian curvature (Section 3) predicts that PCA conflict directions should *partially* align with the top Hessian eigenvectors. We test this on two MLLMs spanning roughly an order of magnitude in trainable-parameter scale: Qwen2.5-VL-3B (Bai et al., 2025) ($d \approx 3$B) and SmolVLM-256M (Marafioti et al., 2025) ($d \approx 256$M).

**Setup.** We use a four-stage protocol per model. *(i) Task-level gradients.* From Vision-FLAN tasks with $\geq 50$ samples we sample 136 tasks (seed=42) and compute a per-task SFT gradient over a fixed set of 50 examples per task; the same example indices are reused in stage (iii). *(ii) PCA-based task selection.* We form the $136\times136$ cosine-similarity matrix, double-center it, embed via the top-2 eigenvectors, and pick the task farthest from the origin in each PC1/PC2-sign quadrant, giving 4 representative tasks. *(ii.5) Conflict directions.* On the $4\times4$ Gram matrix $G_{ij}=\langle g_i, g_j\rangle$ we take the top-3 eigenvectors and lift them to $\mathbb{R}^d$, for both the raw Gram and its double-centered variant. *(iii) Hessian top eigenvectors.* On the $4\times50=200$ examples from the selected tasks (reusing the stage (i) indices) we run 50 Lanczos iterations with full reorthogonalization (fp32 central-difference Hessian–vector products, $\varepsilon=0.1$), yielding the top-10 Hessian eigenvectors at $\theta^{(0)}$. *(iv) Alignment.* We report $|\cos(u_i, h_j)|$, principal angles, and an alignment $z$-score against a Gaussian random-vector baseline, for both constructions.

**Results.** Table 21 reports exactly the alignment our theory predicts. The overlap between PCA conflict directions and the top-10 Hessian eigenvectors is *partial* in absolute terms—leading principal angles of 55–70° and max $|\cos|$ of 0.30–0.48—matching the claim that gradient-PCA identifies high-curvature axes *preferentially*, not perfectly. That this partial overlap is nonetheless far from random is confirmed by the Gaussian-baseline $z$-scores ($\geq 10^4$ at both scales), which rule out chance alignment but do not by themselves imply tight alignment. SmolVLM-256M aligns more strongly than Qwen2.5-VL-3B (top-1 mean $|\cos|$ is $2.4\times$ larger, leading principal angle $14°$ smaller), consistent with curvature concentrating along fewer directions at smaller scale. The double-centered variant tracks the raw one closely.

### F.4. Formal Justification of Gradient–Curvature Alignment in a Tractable Setting

This section proves the splitting-optimality claim (Section 3, Proposition 3.2) in a tractable linear-quadratic model. The model is simplified but captures the essential mechanism: gradient conflict at a shared initialization is *curvature-coupled*, so PCA on gradient similarities naturally finds the directions where merging gains the most.

**Setup.** Consider $T$ datasets with quadratic losses sharing a Hessian,

$$L_t(\theta) = \tfrac{1}{2} (\theta - \theta_t^\star)^\top H (\theta - \theta_t^\star), \qquad t = 1, \ldots, T,$$

where $H = \mathrm{diag}(\lambda_1, \ldots, \lambda_d)$, $\lambda_1 \geq \cdots \geq \lambda_d > 0$. The joint optimum is $\bar{\theta}^\star = \frac{1}{T} \sum_t \theta_t^\star$, and we initialize at $\theta^{(0)} = \bar{\theta}^\star$ so the mean gradient vanishes. The gradient of dataset $t$ at $\theta^{(0)}$ is

$$g_t = \nabla L_t(\theta^{(0)}) = -H \Delta_t, \qquad \Delta_t := \theta_t^\star - \bar{\theta}^\star, \quad \sum_t \Delta_t = 0.$$

**Key identity: gradients are curvature-amplified.** The gradient covariance is

$$\Sigma_g := \frac{1}{T} \sum_t g_t g_t^\top = H \Sigma_\theta H, \qquad \Sigma_\theta := \frac{1}{T} \sum_t \Delta_t \Delta_t^\top.$$

*Remark* F.5 (Curvature amplification). Since $\Sigma_g = H \Sigma_\theta H$, gradient-PCA is steered not by the optima spread $\Sigma_\theta$ alone but by $H \Sigma_\theta H$: in the diagonal case the gradient variance along coordinate $j$ is $\lambda_j^2 \mathrm{Var}(\Delta_{t,j})$. Directions that are both high-curvature and high-disagreement are amplified quadratically, so PCA on gradient conflicts preferentially surfaces high-curvature axes.

**Fine-tuning approximation.** For a balanced split into $G_1, G_2$ ($|G_1|=|G_2|=T/2$), a single-step approximation gives $\theta_k \approx \theta^{(0)} - \eta \bar{g}_k$ with $\bar{g}_k = \frac{1}{|G_k|} \sum_{t \in G_k} g_t$ and displacement $\delta_k = -\eta \bar{g}_k$. The merging gain (Theorem 3.1) for $w_1 = w_2 = 1/2$ is

$$\mathcal{G}_{\mathrm{var}} = \tfrac{1}{8} (\delta_1 - \delta_2)^\top H (\delta_1 - \delta_2) = \tfrac{\eta^2}{8} (\bar{g}_1 - \bar{g}_2)^\top H (\bar{g}_1 - \bar{g}_2). \tag{7}$$

**Partition encoding.** Encode a balanced split by signs $s \in \{\pm 1\}^T$ with $\sum_t s_t = 0$ ($s_t = +1$ if $t \in G_1$). Then $\bar{g}_1 - \bar{g}_2 = \frac{2}{T} \sum_t s_t g_t$ and

$$\mathcal{G}_{\mathrm{var}}(s) = \tfrac{\eta^2}{2T^2} \mathbf{s}^\top M \mathbf{s}, \qquad M_{ij} := g_i^\top H g_j = \Delta_i^\top H^3 \Delta_j,$$

so the gain is governed by the $H^3$-weighted gradient Gram matrix $M = GHG^\top$.

**Concrete example** ($T=4$, $d=2$). Let $H = \mathrm{diag}(\lambda_1, \lambda_2)$ with $\lambda_1 > \lambda_2 > 0$ and centered optima $\Delta_1 = (\alpha, \beta)$, $\Delta_2 = (-\alpha, \beta)$, $\Delta_3 = (\alpha, -\beta)$, $\Delta_4 = (-\alpha, -\beta)$ $(\alpha, \beta > 0)$, so $g_t = -H\Delta_t$. The three distinct balanced partitions give:

- $\mathcal{P}_1$ (*PCA-aligned*, split along coord. 1): $\bar{g}_1 - \bar{g}_2 = (-2\lambda_1\alpha, 0)$, $\mathcal{G}_{\mathrm{var}}^{(1)} = \tfrac{\eta^2}{2} \lambda_1^3 \alpha^2$.

- $\mathcal{P}_2$ (*orthogonal*, split along coord. 2): $\bar{g}_1 - \bar{g}_2 = (0, -2\lambda_2\beta)$, $\mathcal{G}_{\mathrm{var}}^{(2)} = \tfrac{\eta^2}{2} \lambda_2^3 \beta^2$.

- $\mathcal{P}_3$ (*canceling*): $\bar{g}_1 - \bar{g}_2 = (0, 0)$, $\mathcal{G}_{\mathrm{var}}^{(3)} = 0$.

The gradient Gram matrix has coordinate variances $\lambda_1^2 \alpha^2$ and $\lambda_2^2 \beta^2$, so its top PCA direction is coordinate 1 exactly when

$$\lambda_1^2 \alpha^2 > \lambda_2^2 \beta^2, \tag{8}$$

under which PCA selects $\mathcal{P}_1$.

**Proposition F.6** (PCA-aligned splitting maximizes $\mathcal{G}_{\mathrm{var}}$). *Under* (8), *$\mathcal{P}_1$ attains the maximum gain among all balanced partitions:* $\mathcal{G}_{\mathrm{var}}^{(1)} = \tfrac{\eta^2}{2} \lambda_1^3 \alpha^2 \geq \tfrac{\eta^2}{2} \lambda_2^3 \beta^2 = \mathcal{G}_{\mathrm{var}}^{(2)} > 0 = \mathcal{G}_{\mathrm{var}}^{(3)}$.

*Proof.* It suffices that $\lambda_1^3 \alpha^2 \geq \lambda_2^3 \beta^2$ under (8), which gives $\alpha^2/\beta^2 > (\lambda_2/\lambda_1)^2$. Then $\frac{\lambda_1^3 \alpha^2}{\lambda_2^3 \beta^2} = (\lambda_1/\lambda_2)^3 (\alpha^2/\beta^2) > (\lambda_1/\lambda_2)^3 (\lambda_2/\lambda_1)^2 = \lambda_1/\lambda_2 \geq 1$. $\square$

**Corollary F.7** (PCA split dominates random split in expectation). *A uniformly random balanced partition has expected gain* $\mathbb{E}[\mathcal{G}_{\text{var}}^{\text{rand}}] = \frac{\eta^2}{6}(\lambda_1^3\alpha^2 + \lambda_2^3\beta^2) < \frac{\eta^2}{2}\lambda_1^3\alpha^2 = \mathcal{G}_{\text{var}}^{\text{PCA}}.$

*Proof.* Averaging the three equally likely partitions gives $\frac{1}{3}(\mathcal{G}_{\text{var}}^{(1)} + \mathcal{G}_{\text{var}}^{(2)} + \mathcal{G}_{\text{var}}^{(3)})$. The strict inequality holds whenever $\lambda_1^3\alpha^2 > \frac{1}{2}\lambda_2^3\beta^2$, guaranteed by Proposition F.6. $\qquad\square$

*Remark* F.8 (Why the $\lambda^3$ scaling). The PCA-aligned split concentrates variance along $\lambda_1$ for gain $\propto \lambda_1^3\alpha^2$; the orthogonal split yields $\propto \lambda_2^3\beta^2$; the canceling split yields 0. The $\lambda^3$ scaling appears because curvature enters three times—twice through the gradient conflict ($g_t \propto H\Delta_t$, giving $H^2$) and once through the gain formula ($\mathcal{G}_{\text{var}} \propto \delta^\top H\delta$, giving $H^1$). This is why the advantage of conflict-aware splitting grows with the curvature gap $\lambda_1/\lambda_2$.

*Remark* F.9 (Gradient PCA is a safe proxy). The $\mathcal{G}_{\text{var}}$-optimal split is set by $M = GHG^\top$ (eigenvalue weight $\lambda_j^3$), while gradient PCA uses $K = GG^\top$ (weight $\lambda_j^2$). Since $\lambda^3$ is even more biased toward high curvature than $\lambda^2$, whenever gradient PCA picks the high-curvature axis the $\mathcal{G}_{\text{var}}$-optimal split agrees ($\lambda_1^2\alpha^2 > \lambda_2^2\beta^2 \Rightarrow \lambda_1^3\alpha^2 > \lambda_2^3\beta^2$). Gradient PCA is thus a sufficient (if not strictly necessary) criterion for the gain-maximizing split, which is why MERIT can partition on gradient similarities alone.

**General result for $T$ datasets.** We now move beyond $T{=}4$ to compare PCA-aligned and random partitioning for general $T$. Retain the quadratic setting with $M_{ij} = \Delta_i^\top H^3 \Delta_j$. Since $\sum_t g_t = 0$, $M\mathbf{1} = 0$, so 0 is an eigenvalue with eigenvector $\mathbf{1}/\sqrt{T}$; let $\mu_1 \geq \cdots \geq \mu_{T-1} > 0 = \mu_T$ be the eigenvalues with unit eigenvectors $v_1, \ldots, v_T$. For a balanced split $\mathbf{s}$, $\mathcal{G}_{\text{var}}(\mathbf{s}) = \frac{\eta^2}{2T^2} \mathbf{s}^\top M \mathbf{s}.$

**Proposition F.10** (Expected gain under random partitioning). *For* $\mathbf{s}$ *drawn uniformly from balanced sign vectors,* $\mathbb{E}[\mathcal{G}_{\text{var}}^{\text{rand}}] = \frac{\eta^2}{2T(T-1)} \operatorname{tr}(M) = \frac{\eta^2}{2T(T-1)} \sum_{k=1}^{T-1} \mu_k.$

*Proof.* For a uniform balanced sign vector, $\mathbb{E}[s_i^2] = 1$ and $\mathbb{E}[s_i s_j] = -\frac{1}{T-1}$ ($i \neq j$). Hence $\mathbb{E}[\mathbf{s}^\top M\mathbf{s}] = \operatorname{tr}(M) - \frac{1}{T-1}(\mathbf{1}^\top M\mathbf{1} - \operatorname{tr}(M)) = \frac{T}{T-1}\operatorname{tr}(M)$, using $M\mathbf{1} = 0$. Dividing by $2T^2/\eta^2$ gives the result. $\qquad\square$

**Proposition F.11** (PCA-aligned splitting dominates random splitting). *Let* $\mathbf{s}^* = \operatorname{sign}(v_1)$, *where* $v_1$ *is the leading eigenvector of* $M$ *(adjusted for balance if needed[1]). Then*

$$\mathcal{G}_{\text{var}}^{\text{PCA}} = \frac{\eta^2}{2T^2} (\mathbf{s}^*)^\top M \mathbf{s}^* \geq \frac{\eta^2}{2T^2} \mu_1 \|v_1\|_1^2, \tag{9}$$

*where* $\|v_1\|_1 = \sum_t |v_{1,t}|$. *Consequently* $\mathcal{G}_{\text{var}}^{\text{PCA}} > \mathbb{E}[\mathcal{G}_{\text{var}}^{\text{rand}}]$ *whenever*

$$\mu_1 \|v_1\|_1^2 > \frac{T}{T-1} \operatorname{tr}(M). \tag{10}$$

*Proof.* Decompose $\mathbf{s}^* = \sum_k (\mathbf{s}^{*\top} v_k) v_k$, so $(\mathbf{s}^*)^\top M\mathbf{s}^* = \sum_k \mu_k(\mathbf{s}^{*\top} v_k)^2 \geq \mu_1(\mathbf{s}^{*\top} v_1)^2$. With $\mathbf{s}^* = \operatorname{sign}(v_1)$, $\mathbf{s}^{*\top} v_1 = \sum_t |v_{1,t}| = \|v_1\|_1$, giving (9); comparing with Proposition F.10 yields (10). $\qquad\square$

The dominance condition (10) asks for two things. First, spectral concentration: $\mu_1$ should be a large fraction of $\operatorname{tr}(M) = \sum_k \mu_k$, i.e., the conflict structure has a dominant axis. This is exactly the regime where structured splitting helps, whereas isotropic conflicts leave no room over random. Second, eigenvector delocalization: $\|v_1\|_1$ should be large ($1 \leq \|v_1\|_1 \leq \sqrt{T}$, with $\sqrt{T}$ when entries are equal-magnitude), meaning many datasets contribute to the dominant axis, a mild requirement for large mixtures.

**Corollary F.12** (Limiting regimes). *Under Proposition F.11:*

1. **Rank-1 conflicts:** *if* $M \approx \mu_1 v_1 v_1^\top$ *and* $v_1$ *is maximally spread* ($|v_{1,t}| = 1/\sqrt{T}$), *then* $\mathcal{G}_{\text{var}}^{\text{PCA}}/\mathbb{E}[\mathcal{G}_{\text{var}}^{\text{rand}}] \geq T - 1$: *PCA captures all conflict structure while random wastes a* $(1 - 1/T)$ *fraction.*

2. **Isotropic conflicts:** *if* $\mu_1 = \cdots = \mu_{T-1} = \operatorname{tr}(M)/(T-1)$, *then* (10) *is tight and no strategy beats random—when every direction matters equally, there is no preferred axis to split along.*

---

[1]When $T$ is even and $v_1$ has no zero entries, $\operatorname{sign}(v_1)$ is balanced w.h.p. for generic $v_1$; otherwise we assume balance or reassign at most one dataset, in which case (9) holds up to a single-entry correction.

*Proof.* (1) With $\mathrm{tr}(M) = \mu_1$ and $\|v_1\|_1^2 = T$: $\mathcal{G}_{\mathrm{var}}^{\mathrm{PCA}} \geq \eta^2 \mu_1/(2T)$ and $\mathbb{E}[\mathcal{G}_{\mathrm{var}}^{\mathrm{rand}}] = \eta^2 \mu_1/(2T(T-1))$, giving ratio $\geq T-1$. (2) With uniform eigenvalues every balanced split has the same expected projection onto each eigenspace, so structure gives no advantage. $\square$

Real multimodal conflicts are far from isotropic: the 2D PCA projection in Figure 6 shows a few principal components separating the branch-wise updates along well-defined axes. This places MERIT in the spectral-concentration regime of Corollary F.12(1), where PCA-aligned splitting helps the most.

*Remark* F.13 (Recursive splitting along orthogonal PCA axes). MERIT splits recursively along the top-$r$ (orthogonal) PCA eigenvectors $\{u_j\}_{j=1}^r$ of $C$. In the linear-quadratic model, dispersions from successive splits lie on disjoint eigendirections, so cross-axis contributions to $\mathbf{s}^\top M \mathbf{s}$ vanish and the gain decomposes additively:

$$\mathcal{G}_{\mathrm{var}}^{(r)} \propto \sum_{j=1}^r \lambda_j^3 \alpha_j^2,$$

where $\alpha_j$ is the group-mean dispersion along $u_j$. Each PCA dimension adds a non-negative, cubically curvature-weighted term, predicting the monotone 1D $\rightarrow$ 2D $\rightarrow$ 3D gains reported in Section 5.

## F.5. Implicit Regularization via Averaging-Induced Contraction

This section analyzes the implicit regularization from averaging, beyond the variance reduction $\mathcal{G}_{\mathrm{var}}$. Empirical validation (displacement contraction, perturbation robustness, train-loss vs. generalization gap, linear mode connectivity) is consolidated in Appendix B.

**Displacement decomposition.** With displacements $\delta_i := \theta_i - \theta^{(0)}$ and $\bar{\delta} := \bar{\theta} - \theta^{(0)}$ (recall $\theta^\star = \theta^{(0)}$ in our analysis, so these coincide with the $\delta_i$ of Appendix F.1), the parallel-axis identity gives, exactly,

$$\frac{1}{K} \sum_{i=1}^K \|\delta_i\|^2 = \|\bar{\delta}\|^2 + \frac{1}{K} \sum_{i=1}^K \|\delta_i - \bar{\delta}\|^2. \tag{11}$$

So unless all checkpoints coincide, averaging gives a strictly smaller squared displacement, $\|\bar{\theta} - \theta^{(0)}\| \leq \sqrt{\frac{1}{K} \sum_i \|\theta_i - \theta^{(0)}\|^2}$: merging contracts the model toward the shared initialization.

**Empirical validation.** Displacement contraction, perturbation robustness, and the train-loss vs. generalization gap are verified in Appendix B (Tables 6–7 and Figure 4).

**One mechanism, two regularizers.** Both the quadratic gain $\mathcal{G}_{\mathrm{var}}$ (Theorem F.2) and the contraction (11) are instances of the same identity $\frac{1}{K} \sum_i \| \cdot \|^2 = \|\mathrm{mean}\|^2 + \mathrm{variance}$, applied under the $H$-induced norm (loss along high-curvature directions) and under the Euclidean norm (displacement from initialization), respectively. Averaging reduces variance under both, giving MERIT spectral filtering and norm contraction at once.

**Connection to PAC-Bayes.** Many PAC-Bayes bounds for deterministic predictors scale with the squared distance to a reference prior, often the pretrained initialization. Under a Gaussian prior centered at $\theta^{(0)}$, the complexity term is $\mathrm{KL}(Q \| P) \propto \|\theta - \theta^{(0)}\|^2$. Since the merged model stays strictly closer to $\theta^{(0)}$ than the jointly trained one (Table 6), $\|\bar{\theta} - \theta^{(0)}\| < \|\theta_{\mathrm{joint}} - \theta^{(0)}\|$, it incurs a strictly smaller distance-based complexity term, all else equal.

