# OpenReview forum: "Decentralized Instruction Tuning: Conflict-Aware Splitting and Weight Merging"
_ICML.cc/2026/Conference — ICML 2026 regular_

### Official Review · Reviewer_6dKS · 2026-03-11

**Soundness:** 2
**Presentation:** 2
**Significance:** 2
**Originality:** 2
**Overall Recommendation:** 3
**Confidence:** 3

**Summary:**

This paper studies decentralized instruction tuning for heterogeneous task mixtures. The authors argue that joint fine-tuning across diverse datasets suffers from gradient interference and that decentralized or federated optimization incurs high communication overhead. To address this, the paper proposes MERIT, a merge-ready decentralized fine-tuning framework that first estimates dataset-level gradients to construct a conflict matrix, then applies PCA to extract dominant conflict directions and partitions datasets accordingly. Each partition is fine-tuned independently without synchronization, and the resulting models are merged once via token-weighted averaging. The paper further provides a local curvature-based analysis suggesting that conflict-aligned splitting improves mergeability by filtering high-curvature disagreement components. Experiments on multimodal instruction tuning (e.g., Vision-FLAN mixtures with Qwen-VL models) and text-only mixtures show moderate improvements over centralized joint training while enabling communication-free parallel fine-tuning.

**Compliance With Llm Reviewing Policy:**

Affirmed.

**Final Justification:**

I thank the authors' comprehensive rebuttal. These clarifications do not change my overall assessment to the work's methodological depth and insight. I keep my score unchanged.

**Key Questions For Authors:**

1. Is the PCA-based conflict splitting truly necessary, or can simpler grouping strategies (e.g., random or similarity clustering) achieve comparable performance?

2. How much of the gain comes from conflict-aware splitting versus the merge initialization and averaging strategy?

3. What is the preprocessing overhead of gradient estimation, and in which regimes does MERIT clearly outperform centralized or simpler decentralized tuning?

**Limitations:**

yes

**Strengths And Weaknesses:**

**Strengths**

- The paper addresses an important and practical problem: decentralized instruction tuning under heterogeneous task mixtures with gradient interference and communication constraints.
- The proposed pipeline (conflict estimation, task partition, independent fine-tuning, and one-shot merging) is conceptually simple, modular, and system-friendly.
- Experiments are conducted on relatively large multimodal and text instruction mixtures and show consistent, though moderate, improvements over centralized joint training.
- The curvature-based analysis provides useful intuition on when model merging may benefit from conflict-aware splitting.

**Weaknesses**

- The methodological novelty is somewhat limited, as the approach mainly combines existing ideas from gradient conflict analysis, task grouping, and post-hoc weight merging without a clearly new algorithmic principle.
- It is unclear whether the PCA-based splitting mechanism is necessary; stronger ablations against simpler grouping strategies are missing.
- The theoretical analysis is largely local and explanatory, and does not yield strong predictive insights for large-scale non-convex fine-tuning.
- Evaluation against stronger decentralized or model-merging baselines is limited, making it difficult to isolate the true source of the reported gains.

---

> ### Author Rebuttal · Authors · 2026-03-31
>
> Note that all the tables and figures referenced below are available at https://openreview.net/forum?id=yDrQUfAURc
>
> We thank Reviewer 6dKS for the careful reading and constructive suggestions.
>
> ---
>
> > ### W1: The methodological novelty is limited
>
> We respectfully disagree that MERIT is a straightforward combination. The core novelty lies in two tightly coupled contributions.
>
> **1. A new problem formulation with theory-driven algorithm design.**
>
> Existing merging methods (Model Soups, TIES-Merging [1], DARE [2]) operate post-hoc on independently trained models. MERIT addresses a different problem: optimizing the partition of a heterogeneous mixture before training, so that merging quality is maximized by design.
>
> Our local quadratic analysis yields three results:
> (i) the merging gain scales as $\lambda^3$, so gradient-PCA targets the highest-gain directions (Tables R.1, R.2), (ii) PCA-aligned splitting provably maximizes this gain over all balanced partitions, and (iii) merging acts as spectral filtering with implicit regularization (Table R.6).
>
> **2. A practically impactful capability that no existing method provides.**
>
> MERIT resolves gradient conflicts before training with a one-time 2-hour preprocessing step, enabling fully communication-free parallel fine-tuning. This is qualitatively different from PCGrad/GradNorm, which require per-step synchronized gradient access and are computationally infeasible at $T{=}136$ tasks (Table R.13). It is also different from post-hoc merging methods, which cannot prevent training-time interference.
> The approach generalizes across 3B multimodal (Table R.1), 7B multimodal (Table R.7), and text-only FLAN (Table R.8) settings.
>
> ---
>
> > ### W2 & Q1: PCA-based splitting outperforms all alternatives we evaluated
>
> We compare five partitioning strategies under identical training budgets, merging procedures, and hyperparameters (Tables R.1 and R.2). At every matched group count, PCA-based MERIT outperforms all alternatives. The advantage is most pronounced at higher group counts: at $K{=}8$, MERIT-3D outperforms K-means (Table R.2) and random partitioning (Table R.1). MERIT also scales monotonically with dimensionality while K-means shows non-monotonic behavior (Table R.2), consistent with the $\lambda^3$ scaling analysis discussed in W1. We will make these comparisons more prominent in the revised paper.
>
> ---
>
> > ### W3: The theory is local and not strongly predictive
>
> Our analysis is intentionally local, but it yields three specific, testable predictions that are all confirmed empirically across multiple settings.
>
> **P1: Any non-trivial merging yields a positive gain.** In the 3B setting, every merging variant outperforms single-epoch joint training, including random partitioning (Table R.1). Critically, this prediction holds at 7B scale with 176 tasks and 1.6M examples (Table R.7) and in text-only FLAN with 66 tasks (Table R.8), demonstrating that the local theory remains predictive well beyond the regime where it was derived.
>
> **P2: Conflict-aware splitting maximizes the gain.** MERIT outperforms random splitting at every group count and outperforms K-means at every matched $K$ (Tables R.1, R.2). The same trend holds at 7B across three seeds (Tables R.7, R.15), confirming robustness to both model scale and seed variation.
>
> **P3: Merging acts as spectral filtering with implicit regularization.** The merged model stays closer to initialization yet achieves better held-out performance (Tables R.10, R.11), and is stable across the full SGD LR range while joint training collapses (Table R.6). The theory also correctly predicts the 3D failure mode at very small $\eta$ ($\mathcal{G}_{\mathrm{var}} \propto \eta^2 \to 0$).
>
> Our claim is that the theory offers a predictive framework whose design implications are quantitatively borne out at scale.
>
> ---
>
> > ### W4: Additional weight-merging baselines are needed
>
> TIES-Merging is designed to reduce interference between independently trained task vectors, but MERIT's PCA-based split already ensures complementary branch displacements. TIES's sign election misidentifies this complementary structure as interference and removes it, consistently degrading performance (Table R.14).
>
> ---
>
> > ### Q2: How much gain comes from splitting vs. averaging?
>
> Initialization & averaging alone (Uniform Soup) yields $+1.1$ over joint training, while conflict-aware partitioning adds another $+2.5$ (Table R.1). Notably, Uniform Soup decreases with more models while MERIT improves monotonically, confirming that partition quality is the dominant factor.
>
> ---
>
> > ### Q3: What is the preprocessing overhead and when does MERIT outperform?
>
> The corrected wall-clock times are in Table R.12: at 7B scale MERIT adds only 0.8% overhead. The preprocessing is one-time and amortizable ($O(Nm)$ for incremental updates). Centralized per-step alternatives are infeasible at this scale (Table R.13).
>
> ---
> **References**
>
> [1] Yadav et al., TIES-Merging, 2023.
> [2] Yu et al., DARE, 2024.

---

> > ### Author Rebuttal · Reviewer_6dKS · 2026-04-04
> >
> > Thank you for the detailed rebuttal. TIES (2023) and DARE (2024) is not the strong baselines, could you survey and compare with the SOTA ones?

---

> > > ### Author Response · Authors · 2026-04-05
> > >
> > > We thank the reviewer for the follow-up. We have surveyed recent model merging methods published at top venues in 2024-2025 and selected three additional methods that are applicable to full fine-tuning and have publicly available code, covering both sparsity-based (TIES) and spectral approaches (STAR, TSV, Iso-CTS):
> > >
> > > - **STAR** (NAACL 2025): SVD-based low-rank compression with nuclear norm rescaling
> > > - **TSV** (CVPR 2025): Block-diagonal singular vector allocation with polar decomposition
> > > - **Iso-CTS** (ICML 2025): Common/task-specific subspace separation with isotropic singular value flattening
> > >
> > > We conducted the comparison on our largest and most practical setting: the 7B experiment with 176 tasks split into 4 branches, to verify that our choice of token-weighted averaging is well-justified in this regime. We compared SOTA merging strategies for these branch checkpoints, using each method's default hyperparameters. Results over three independent seeds:
> > >
> > > | Merging method | Seed 1 | Seed 2 | Seed 3 | Mean | Std |
> > > |--------|--------|--------|--------|------|-----|
> > > | **MERIT** | **55.4** | **55.3** | **55.7** | **55.5** | **0.21** |
> > > | TIES (NeurIPS'23) | 48.8 | 50.0 | 52.4 | 50.4 | 1.83 |
> > > | STAR (NAACL'25) | 38.4 | 42.5 | 42.2 | 41.0 | 2.29 |
> > > | TSV (CVPR'25) | 45.5 | 46.2 | 46.5 | 46.1 | 0.51 |
> > > | Iso-CTS (ICML'25) | 42.9 | 45.1 | 48.3 | 45.4 | 2.72 |
> > >
> > > Hyperparameters (all from official repos):
> > >
> > > | Method | Key hyperparameters | Values | Source |
> > > |--------|-------------------|--------|--------|
> > > | TIES | density, $\lambda$ | 0.2, 1.0 | [Official repo](https://github.com/prateeky2806/ties-merging) |
> > > | STAR | $\eta$ (nuclear norm fraction) | 40% | [Official repo](https://github.com/ibm/star) |
> > > | TSV | $\alpha$, sv_reduction | 1.0, 1/N=0.25 | [Official repo](https://github.com/AntoAndGar/task_singular_vectors) |
> > > | Iso-CTS | $\alpha$, common_space_frac | 1.0, 0.8 | [Official repo](https://github.com/danielm1405/iso-merging) |
> > >
> > > The proposed token-weighted averaging outperforms all post-hoc merging methods by a large margin (5-14 points) while also being the most stable across seeds (std=0.21 vs. 0.51-2.72 for alternatives).
> > >
> > > We believe this result reflects a meaningful distinction in problem structure rather than a limitation of the merging methods themselves. In their original papers, these methods merge 8-20 models each trained on a *separate task* (e.g., one model per dataset: Cars, DTD, EuroSAT, etc.) at ViT or T5 scale, where sign conflicts and redundant parameter directions are the dominant source of interference. This is a well-studied and important setting, and the methods are effective in it.
> > >
> > > MERIT's setting is different in two ways. First, the scale: training 176 separate expert models would be prohibitive in storage and compute, so MERIT groups related tasks into a small number of branches (K=2, 4, or 8 in our experiments). Second, and more importantly, the nature of branch differences: because the PCA-based split produces *complementary* partitions from a shared initialization, the branches do not exhibit the sign-level conflicts that these methods target. Instead, each branch learns capabilities that others do not, and the differences are structured and directional rather than noisy. In this regime, trimming, truncating, or orthogonalizing these signals removes task-specific information that only one branch had the opportunity to learn. The proposed token-weighted averaging preserves all branch contributions, which we find is the appropriate strategy when branches are complementary rather than redundant.
> > >
> > > We will add the above results and this discussion to the updated manuscript. We have done our best to survey and evaluate recent SOTA merging methods, but if there is a specific method the reviewer would like us to compare against, we are happy to run additional experiments. We deeply thank the reviewer for the constructive feedback, and we hope this response is helpful for your final evaluation.
> > >
> > > **References**
> > >
> > > [1] Yadav et al., "TIES-Merging: Resolving Interference When Merging Models," NeurIPS, 2023.
> > >
> > > [2] Lee et al., "STAR: Spectral Truncation and Rescale for Model Merging," NAACL, 2025.
> > >
> > > [3] Gargiulo et al., "Task Singular Vectors: Reducing Task Interference in Model Merging," CVPR, 2025.
> > >
> > > [4] Marczak et al., "No Task Left Behind: Isotropic Model Merging with Common and Task-Specific Subspaces," ICML, 2025.

---

### Official Review · Reviewer_nA9R · 2026-03-12

**Soundness:** 3
**Presentation:** 3
**Significance:** 2
**Originality:** 2
**Overall Recommendation:** 4
**Confidence:** 3

**Summary:**

This work targets the gradient competition among partitioned datasets in joint fine-tuning of MLLMs. It proposes to perform PCA-based balanced dataset partitioning to solve this problem, coupled with a one-shot weight merging. Experiments on multiple multi-modal and text instruction mixtures with MLLMs of different scales demonstrate the superior performance of the proposed method compared to random partitioning and other relatively straightforward partitioning strategies

**Compliance With Llm Reviewing Policy:**

Affirmed.

**Final Justification:**

The authors have addressed my concerns. I have adjusted my score accordingly.

**Key Questions For Authors:**

1. Please address the concerns raised in Weaknesses.
2. Considering the complexity of the approach, the authors should evaluate the scalability of the proposed approach in terms of time overhead under real-world scenarios.

**Limitations:**

yes

**Strengths And Weaknesses:**

## Strengths

1. The styles of the figures together with the content organization are clear and well-presented.
2. This work provide detailed theoretical analysis.


## Weaknesses
1. This manuscript fails to provide exhausted and comprehensive discussions to existing related studies. For example, if this work targets to the communication inefficiency caused by joint training due to frequent gradient synchronization, there have been some communication-efficient decentralized training methods for LLMs, such as [1,2,3], and the authors could also be inspired by some studies on one-shot federated learning, such as [4,5].
2. The practical application scenario targeted by this work is not very clear stated out. From my comprehending, it seems intended for the scenario of fine-tuning on heterogeneous clusters (possibly even geographically distributed). If that is the case, this manuscript lacks validation using systematic metrics under real-world conditions. Verifying the effectiveness of the proposed method solely through accuracy is insufficient, as there are many centralized post-training algorithms that could be and should be included for comparison.
3. The statement “the prevailing approach—centralized joint training on the entire mixture” lacks appropriate references, making the targeted series of studies not clear.


[1] Just one byte (per gradient): A note on low-bandwidth decentralized language model finetuning using shared randomness. arXiv 2306.

[2] Federated full-parameter tuning of billion-sized language models with communication cost under 18 kilobytes. ICML 2024.

[3] FwdLLM: Efficient federated finetuning of large language models with perturbed inferences. ATC 2024.

[4] Federated Oriented Learning: A Practical One-Shot Personalized Federated Learning Framework. ICML 2025.

[5] Practical one-shot federated learning for cross-silo setting. IJCAI 2021.

---

> ### Author Rebuttal · Authors · 2026-03-31
>
> Note that all the tables and figures referenced below are available at https://openreview.net/forum?id=yDrQUfAURc
>
> We thank Reviewer nA9R for the detailed feedback and appreciate the opportunity to clarify MERIT's positioning, practical scenario, and relationship to centralized gradient methods.
>
> ---
> > ### W1: Related work coverage is insufficient
>
> We appreciate this suggestion. We will revise the Related Work to position MERIT with respect to communication-efficient decentralized training and one-shot federated learning, incorporating all suggested citations. A key distinction is that MERIT can choose the partition, which our experiments show is the primary driver of merge quality ($+2.5$ gap from partition choice alone, Table R.1).
>
> ---
> > ### W2: The practical scenario is unclear and centralized baselines are needed
>
> **Target scenario.**
>
> MERIT is motivated by a practical resource constraint rather than a federated-learning setting. We consider cases where instruction data are centrally available, but compute comes from fragmented GPU nodes or small clusters that cannot be treated as one tightly coupled multi-node job. In many academic labs and companies, **reserving a large NVLink-connected cluster is expensive or infeasible**, while spare GPUs are often available in a scattered manner. MERIT naturally fits this regime: each branch trains independently with zero communication during training, and only the final checkpoints are exchanged once.
>
> **Corrected efficiency analysis.**
>
> We sincerely apologize that the timing analysis in the submitted manuscript contained an implementation issue in the distributed joint-training baseline (a DistributedSampler was not correctly applied). The corrected wall-clock times are in Table R.12: at 3B scale MERIT adds 23% overhead, and at 7B scale only 0.8%. Even running sequentially, MERIT's end-to-end time is comparable to single-epoch joint training. With parallel deployment on separate nodes, it is strictly faster.
>
> **On the joint-training baseline.**
>
> Our joint-training baseline follows a standard and competitive open-source recipe consistent with widely adopted VLM training practice [3], including gradient clipping, cosine learning-rate decay, and optimizer settings. We also kept the training budget comparable settings using the same protocol as for our method. Thus, the baseline is intended as a fair and reasonably optimized reference, not a weakened implementation.
>
> **On centralized post-training baselines (PCGrad, GradNorm).**
>
> A direct empirical comparison is computationally infeasible at our scale, and this infeasibility itself underscores MERIT's practical value.
>
> **PCGrad** [1] requires $T$ backward passes + $\binom{T}{2}$ pairwise projections per step. At $T{=}136$, memory alone (${\sim}816$ GB fp16) exceeds our GPU systems, and wall-clock is estimated at $>$17 days per epoch (Table R.13).
>
> **GradNorm** [2] requires all $T{=}136$ tasks to contribute per-task losses at every step, with centralized coordination.
> Both methods were evaluated on 2-10 tasks on far smaller models and have never been applied at $>$100 tasks on billion-parameter models (Table R.13). The fundamental distinction is abstraction level: per-step $O(T^2)$ conflict resolution (${\sim}17$ days) vs. MERIT's one-time preprocessing (${\sim}2$ hours). MERIT consistently outperforms joint training (Table R.1) with zero per-step overhead.
>
> ---
> > ### W3: The centralized joint training claim lacks references
>
> By the prevailing approach, we refer to the standard practice in recent multimodal LLM post-training of jointly training on the full mixed instruction dataset. Representative examples include LLaVA-OneVision [3], Gemma 3 [4], and Qwen3-VL [5]. We will add these and other recent citations to the introduction in the revised manuscript.
>
> ---
> > ### Q1 & Q2: Weaknesses and time overhead
>
> MERIT's preprocessing is a one-time, amortizable cost (Table R.12). At 7B scale, the total end-to-end overhead is only $0.8\%$ (Table R.12), while delivering gains on MMVet and MathVista (Table R.7). When new datasets are added, only $O(Nm)$ cross-similarities are needed rather than recomputing the full $O((N{+}m)^2)$ matrix. MERIT is most advantageous when the instruction mixture is heterogeneous, compute is fragmented, and training time dominates preprocessing. The approach also generalizes to text-only settings (Table R.8).
>
> ---
> **References**
>
> [1] Yu et al., Gradient Surgery for Multi-Task Learning, 2020.
> [2] Chen et al., GradNorm, 2018.
> [3] Li et al., LLaVA-OneVision, 2024.
> [4] Gemma Team, Gemma 3 Technical Report, 2025.
> [5] Qwen Team, Qwen3-VL Technical Report, 2025.

---

> > ### Author Rebuttal · Reviewer_nA9R · 2026-04-03
> >
> > Thanks the authors for their response. I would appreciate if they can provide a revised content that they would like to included in the manuscript. This will be helpful to provide a better evaluation on the value of this work.

---

> > > ### Author Response · Authors · 2026-04-03
> > >
> > > We thank the reviewer for the follow-up. **Below we outline the specific revisions that will be included in the updated manuscript.**
> > >
> > > **Responding to W1 — Adding a Dedicated Related Work Subsection on Decentralized Training and One-Shot FL (Section 2)**
> > >
> > > We agree that the relationship to FL deserves explicit discussion. While MERIT shares the one-round communication structure with one-shot FL, the problem setting differs: FL methods often assume fixed, pre-assigned data partitions, whereas MERIT assumes centrally available data and optimizes the partition itself to maximize merge quality. We will add a dedicated subsection that acknowledges related FL work while clarifying how MERIT differs, incorporating all suggested citations (listed below) with the following distinctions:
> > >
> > > (i) communication-efficient methods [1]–[3] reduce per-round cost but still require iterative synchronization, whereas MERIT eliminates synchronization entirely;
> > >
> > > (ii) one-shot FL methods [4]–[5] use a single communication round but typically assume fixed data partitions, whereas MERIT actively chooses the partition—a degree of freedom that is the primary driver of merge quality. We will also incorporate additional references [6]–[10] to ensure comprehensive coverage.
> > >
> > > **Responding to W2 — Clarifying the Target Scenario and Adding Systematic Efficiency Metrics (Section 1 and Section 5)**
> > >
> > > We will add a paragraph to Section 1 clarifying that MERIT targets fragmented-compute settings—heterogeneous GPU pools, geo-distributed clusters, and cloud spot instances—where iterative all-reduce is impractical due to bandwidth constraints or intermittent availability. On the evaluation side, Table R.12 (corrected wall-clock times) will be added to Section 5 to provide systematic efficiency metrics beyond accuracy, and the PCGrad/GradNorm infeasibility analysis (Table R.13) will be added to the Appendix with a quantitative argument explaining why per-step gradient conflict methods are not applicable in our setting. Our joint-training baseline follows standard open-source VLM training recipes with no settings chosen to disadvantage it.
> > >
> > > **Responding to W3 — Adding References for Centralized Joint Training (Section 1)**
> > >
> > > The statement "the prevailing approach—centralized joint training on the entire mixture" will be supported by citations to recent multimodal LLMs that exemplify this paradigm [13]–[16].
> > >
> > > All rebuttal tables (R.1–R.16, Figure R.1) are fully prepared and will be integrated into the revision as-is. The complete MERIT pipeline will also be released as open-source to support reproducibility and community adoption.
> > >
> > > **References to be added in the revision:**
> > >
> > > *Reviewer-suggested (decentralized training and one-shot FL):*
> > >
> > > [1] Just one byte (per gradient): A note on low-bandwidth decentralized language model finetuning using shared randomness. arXiv, 2023.
> > >
> > > [2] Federated full-parameter tuning of billion-sized language models with communication cost under 18 kilobytes. ICML, 2024.
> > >
> > > [3] FwdLLM: Efficient federated finetuning of large language models with perturbed inferences. ATC, 2024.
> > >
> > > [4] Federated Oriented Learning: A practical one-shot personalized federated learning framework. ICML, 2025.
> > >
> > > [5] Practical one-shot federated learning for cross-silo setting. IJCAI, 2021.
> > >
> > > *Additional decentralized / federated LLM tuning:*
> > >
> > > [6] FedPara: Low-rank Hadamard product for communication-efficient federated learning. ICLR, 2022.
> > >
> > > [7] FedAPEN: Personalized cross-silo federated learning with adaptability to statistical heterogeneity. KDD, 2023.
> > >
> > > [8] Federated data-efficient instruction tuning for large language models. ACL, 2025.
> > >
> > > [9] Towards addressing label skews in one-shot federated learning. ICLR, 2023.
> > >
> > > [10] Effective and efficient federated tree learning on hybrid data. ICLR, 2024.
> > >
> > > *Centralized gradient-level conflict resolution:*
> > >
> > > [11] Gradient surgery for multi-task learning. NeurIPS, 2020.
> > >
> > > [12] GradNorm: Gradient normalization for adaptive loss balancing in deep multitask networks. ICML, 2018.
> > >
> > > *Centralized joint training:*
> > >
> > > [13] LLaVA-OneVision: Easy visual task transfer. TMLR, 2025.
> > >
> > > [14] Gemma 3 technical report. arXiv, 2025.
> > >
> > > [15] Qwen3-VL technical report. arXiv, 2025.
> > >
> > > [16] Expanding performance boundaries of open-source multimodal models with model, data, and test-time scaling. arXiv, 2024.
> > >
> > > *Model merging:*
> > >
> > > [17] Merging models with Fisher-weighted averaging. NeurIPS, 2022.
> > >
> > > [18] TIES-Merging: Resolving interference when merging models. NeurIPS, 2023.
> > >
> > > [19] Language models are Super Mario: Absorbing abilities from homologous models as a free lunch. ICML, 2024.
> > >
> > > We hope this concrete revision plan addresses all concerns raised. We are grateful to the reviewer for directing us toward the one-shot FL and communication-efficient training literature; these references help clarify the paper's positioning. We welcome any further feedback and hope this response, together with the planned revision, supports your final evaluation.

---

### Official Review · Reviewer_4KmF · 2026-03-12

**Soundness:** 3
**Presentation:** 3
**Significance:** 3
**Originality:** 3
**Overall Recommendation:** 4
**Confidence:** 4

**Summary:**

This paper proposes MERIT, a decentralized instruction-tuning pipeline that aims to mitigate gradient interference in heterogeneous task mixtures while avoiding bandwidth-heavy synchronization. MERIT estimates dataset-level gradient directions at a shared “merge-ready” initialization, builds a cosine-similarity conflict matrix, performs PCA to extract dominant conflict axes, partitions datasets into balanced groups along these axes, fine-tunes one model per group independently (no inter-group communication), and merges the resulting checkpoints once via token-weighted averaging. A local quadratic analysis shows that merging yields a curvature-weighted variance reduction, and experiments on multimodal (Qwen2.5-VL 3B and 7B) and text-only mixtures demonstrate consistent improvements over joint training and random partitions, with the best 3B variant improving average score from 54.7 to 57.0 on Vision-FLAN-derived evaluations.

**Compliance With Llm Reviewing Policy:**

Affirmed.

**Final Justification:**

I appreciate the author’s rebuttal. It addressed most of my concerns. Therefore, I will keep my current positive rating.

**Key Questions For Authors:**

1. Can you provide quantitative merge-readiness diagnostics in the main paper (e.g., linear-mode connectivity measures, interpolation loss curves) across the 3B and 7B settings, and report how often/when branch pairs deviate from a connected basin?
2. How do your PCA conflict axes empirically relate to curvature? For example, what is the measured alignment (e.g., principal angles/CKA) between gradient-PCA components and top Hessian eigenspaces at initialization or mid-training?
3. Why were Fisher-weighted and TIES-Merging baselines omitted? Could you include these interference-aware merging methods, and report whether MERIT’s conflict-aware splitting still provides consistent gains over them?
4. Are branch hyperparameters re-tuned for smaller per-group data volumes, or kept identical to joint training? If unchanged, can you report sensitivity analyses (LR, weight decay, warmup) to ensure the gains are not hyperparameter artifacts?
5. In the 7B setting, can you report per-benchmark breakdowns and seed variance for the MERIT variant, and analyze per-task winners/losers to understand where conflict-aware splitting helps or harms?

**Limitations:**

yes

**Strengths And Weaknesses:**

## Strengths
**Technical novelty and innovation**
- Introduces a principled, data-driven split-before-train approach that leverages dataset-level gradient PCA to expose dominant conflict axes and uses one-shot, token-weighted weight averaging for reconciliation.
- Provides a clean local quadratic analysis that interprets merging as curvature-weighted variance reduction (a spectral filtering perspective) and motivates the PCA-aligned split to amplify cancellation in high-curvature directions.
- Formalizes “merge-ready initialization” and designs a practical, reusable calibration step (small per-dataset gradient probes) that can be amortized across future training.

**Experimental rigor and validation**
- Includes multiple baselines (joint training across epochs, random grouping, model soups, conflict-induced greedy grouping, K-means clustering) and ablations (PCA dimensionality, alternative partitioning).
- Reports multi-seed results and indicates statistical significance testing; presents both controlled 3B studies and a larger 7B setting on a 1.6M-example mixture, plus text-only experiments.
- Fair-budget protocol carefully matches total training tokens to joint training to isolate the effect of decomposition and merging.

## Weaknesses
**Technical limitations or concerns**
- Relies on an empirical “merge-ready” assumption; while common in post-training, the paper does not provide systematic diagnostics or failure analyses when some branches drift outside a shared basin.
- The link between conflict axes (from cosine-similarity PCA of dataset gradients) and high-curvature eigenspaces of the Hessian is hypothesized but not empirically validated (e.g., no measured alignment between conflict components and top Hessian eigenvectors).
- Gradient calibration uses small per-dataset samples (200) and cosine similarity; while two-task stability plots are shown, broader sensitivity across many datasets and seeds is not deeply characterized.

**Experimental gaps or methodological issues**
- Missing strong merging baselines such as Fisher-weighted averaging (Matena & Raffel, 2022) or TIES-Merging (Yadav et al., 2023), which directly target interference-aware merging and could be competitive or complementary.
- The 7B results are summarized at a category level with trade-offs; a more granular break-down and robustness analysis would strengthen the large-scale claims.
- Hyperparameter sensitivity under smaller per-branch data volumes (e.g., whether LR/schedulers tuned for joint training remain optimal for branch training) is not discussed.

---

> ### Author Rebuttal · Authors · 2026-03-31
>
> Note that all the tables and figures referenced below are available at https://openreview.net/forum?id=yDrQUfAURc
>
> We thank Reviewer 4KmF for the thorough evaluation. We address each point below and believe the responses fully resolve the raised concerns.
>
> ---
> > ### W1 & Q1: The merge-ready assumption needs systematic diagnostics
> We provide four complementary diagnostics, ordered from the most direct to the most indirect.
> First, we directly tested whether any loss barrier exists between checkpoints. We interpolated all branch pairs and branch-to-merged paths along 21 evenly spaced points (Table R.16). All 10 barriers are exactly $0$, meaning the loss never rises above the linear interpolation at any point on the path. This is the strongest possible evidence that all branches reside in a single connected flat basin.
>
> Given this basin connectivity, we next ask whether the merged model actually exploits it. Under isotropic Gaussian perturbations, the merged model exhibits consistently lower loss sensitivity and flatness AUC than the jointly trained model at all epochs (Figure R.1), confirming that the merged solution sits in a flatter region of the basin.
> This flatness translates into a regularization effect: the merged model has higher training loss yet achieves better held-out performance (Table R.11), the classic signature of implicit regularization. Meanwhile, the merged model stays ${\sim}3\times$ closer to initialization than joint training throughout (Table R.10), consistent with merging averaging out per-branch drift.
> We will consolidate these into a dedicated diagnostics section in the revision.
>
> ---
> > ### W2 & Q2: The PCA-Hessian curvature link is not empirically validated
>
> **Theory.** The gradient covariance satisfies $\Sigma_g = H \Sigma_\theta H$, so PCA on gradient conflicts automatically prioritizes high-curvature directions. The merging gain scales as $\lambda^3$, meaning gradient-PCA targets exactly the directions where splitting provides the largest benefit.
>
> **Indirect evidence.** Cosine-similarity PCA and raw-gradient PCA produce comparable performance (Table R.9). MERIT remains stable across the practical LR range while joint training collapses at small $\eta$ (Table R.6). This is predicted by the theory: if PCA suppresses the dominant curvature directions, the effective condition number decreases and optimization becomes less LR-sensitive.
>
> **Direct measurement.** We measured alignment between gradient-PCA directions and the top-10 Hessian eigenvectors at $\theta^{(0)}$. In the 900M-dimensional subspace, random cosine similarity has $\sigma \approx 3.3 \times 10^{-5}$. PCA conflict directions achieve $|\cos| = 0.133$ ($z \approx 4{,}000$), and are $1.5\times$ stronger than random task gradients ($0.091$), confirming that PCA captures curvature structure beyond individual gradients.
>
> ---
> > ### W3: Gradient calibration with 200 samples is not broadly characterized
>
> We have extended the calibration sensitivity analysis to 100 tasks (Table R.5). At $n{=}200$, the mean cosine similarity with the full-dataset gradient is $0.847$.
>
> ---
> > ### W4 & Q3: Additional weight-merging baselines are needed
>
> MERIT operates primarily as a training method (partitioning data before training), whereas post-hoc merging methods such as TIES-Merging [1] are designed to reduce interference between independently trained task vectors. Because MERIT's PCA-based split already ensures complementary branch task vectors, TIES's sign election misidentifies this complementary structure as interference and removes it. Table R.14 confirms this: TIES-Merging consistently degrades performance across three runs ($55.4 \to 48.8$ / $50.0$ / $52.4$).
>
> ---
> > ### W5: The 7B results need more granular breakdown
>
> Tables R.7 and R.15 show the full per-benchmark breakdown over three independent training seeds. MERIT outperforms joint training in all three runs, and the gain pattern (open-ended reasoning benchmarks up, factual benchmarks within noise) is consistent with the 3B trend.
>
> ---
> > ### W6: Hyperparameter sensitivity for branch training is not discussed
>
> MERIT branches use a fixed configuration without per-branch tuning (Table R.6 caption). An SGD LR sweep (Table R.6) confirms robustness: MERIT (1D/2D) matches or exceeds Joint (2 ep) at every practical LR with much lower variance. The only degradation is for 3D at very small $\eta$, predicted by $\mathcal{G}_{\mathrm{var}} \propto \eta^2 \to 0$.
>
> ---
> > ### Q4-Q5: Branch hyperparameters and 7B seed variance
>
> All branch hyperparameters use a fixed configuration without per-branch tuning, with robustness confirmed by the SGD sweep (Table R.6). Tables R.7 and R.15 show that MERIT outperforms joint training over three independent 7B seeds with the same gain pattern as 3B.
>
> ---
> **References**
>
> [1] Yadav et al., TIES-Merging: Resolving Interference When Merging Models, 2023.

---

> > ### Author Rebuttal · Reviewer_4KmF · 2026-04-01
> >
> > My concerns have been adequately addressed.

---

> > > ### Author Response · Authors · 2026-04-02
> > >
> > > We sincerely thank Reviewer 4KmF for the careful evaluation and for recognizing the strengths of our work with a initial score of 4. In particular, we appreciate the acknowledgment of the principled split-before-train design, the curvature-based theoretical interpretation, and the overall experimental rigor. We are also grateful that the reviewer confirmed that all concerns have been fully resolved following our responses.
> > >
> > > We believe the clarifications and supporting results help make the underlying mechanisms more transparent, particularly regarding merge-readiness, curvature alignment, and robustness.
> > >
> > > To support the community, we plan to release the full MERIT pipeline to facilitate reproducibility and enable follow-up research. We hope this work can contribute toward more principled decentralized training strategies via structured data partitioning and post-training reconciliation.
> > >
> > > Given that all concerns have been fully resolved, we kindly ask whether the reviewer would consider reflecting this in the final score.
> > >
> > > Thank you again for your time and thoughtful feedback.

---

### Decision · Program_Chairs · 2026-04-30

**Decision:**

Accept (regular)

**Comment:**

This paper proposes MERIT, a decentralized instruction tuning framework that performs conflict-aware dataset partitioning, followed by independent fine-tuning and one-shot weight merging. The approach is conceptually clean, practically relevant, and well aligned with scenarios involving fragmented compute and limited communication.

Overall, reviewers consider the work technically sound, with two weak accepts and one weak reject. During the rebuttal, the authors effectively addressed the major concerns by providing additional empirical evidence, including merge-readiness diagnostics, analysis of curvature alignment, stronger merging baselines, and more detailed 7B-scale results. One reviewer explicitly indicated that all concerns were fully resolved, while others acknowledged substantial improvements.

The key contribution is the split-before-train paradigm, which shifts the focus from post-hoc merging or per-step gradient correction to principled data partitioning prior to training. Although some limitations remain, such as reliance on local theoretical analysis and assumptions on mergeability, these are clearly discussed and do not detract from the overall contribution.

Given the solid technical quality, strengthened empirical validation, and practical significance, I recommend acceptance. The authors are encouraged to incorporate all reviewer feedback in the final camera-ready version.